# Token Perturbation Guidance for Diffusion Models

**Javad Rajabi**[1,2]  **Soroush Mehraban**[1,2,3]  **Seyedmorteza Sadat**[4]  **Babak Taati**[1,2,3]

[1]University of Toronto    [2]Vector Institute for Artificial Intelligence

[3]KITE Research Institute    [4]ETH Zürich

{rajabi, taati}@cs.toronto.edu
soroush.mehraban@mail.utoronto.ca
seyedmorteza.sadat@inf.ethz.ch

## Abstract

Classifier-free guidance (CFG) has become an essential component of modern diffusion models to enhance both generation quality and alignment with input conditions. However, CFG requires specific training procedures and is limited to conditional generation. To address these limitations, we propose Token Perturbation Guidance (TPG), a novel method that applies perturbation matrices directly to intermediate token representations within the diffusion network. TPG employs a norm-preserving shuffling operation to provide effective and stable guidance signals that improve generation quality without architectural changes. As a result, TPG is training-free and agnostic to input conditions, making it readily applicable to both conditional and unconditional generation. We further analyze the guidance term provided by TPG and show that its effect on sampling more closely resembles CFG compared to existing training-free guidance techniques. Extensive experiments on SDXL and Stable Diffusion 2.1 show that TPG achieves nearly a $2\times$ improvement in FID for unconditional generation over the SDXL baseline, while closely matching CFG in prompt alignment. These results establish TPG as a general, condition-agnostic guidance method that brings CFG-like benefits to a broader class of diffusion models. The code is available at
https://github.com/TaatiTeam/Token-Perturbation-Guidance

## 1   Introduction

Diffusion models [1, 2, 3] have emerged as the main methodology behind many successful generative models for images [4, 5], videos [6, 7, 8, 9, 10], audio [11, 12], and 3D objects [13, 14, 15]. Despite their theoretical capacity to produce high-fidelity data, unguided diffusion models often suffer from poor sample quality, manifesting as visual artifacts, lack of semantic consistency, and insufficient sharp details [16]. To mitigate these issues, classifier-free guidance (CFG) [17] has become the de facto approach to steer the generation process toward higher quality and more semantically aligned outputs. However, CFG is inherently limited to conditional generation and requires a specific training strategy that randomly replaces the input condition with a null condition.

In response to these constraints, several alternative guidance techniques have emerged, aiming to extend CFG-like benefits to broader settings [18, 16, 19, 20, 21]. These approaches often manipulate components of the denoiser network, such as attention layers, to construct effective guidance signals. However, they either require additional specialized training or offer limited improvements in prompt alignment and generation quality (particularly w.r.t. unconditional generation). Accordingly, there remains a need for a training-free guidance mechanism that works across both conditional and unconditional settings while improving *generation quality* and *semantic alignment* similar to CFG.

In this paper, we revisit existing attention-based guidance techniques and aim to bridge the gap between the effectiveness of CFG and that of training-free, condition-agnostic methods. We observe

39th Conference on Neural Information Processing Systems (NeurIPS 2025).

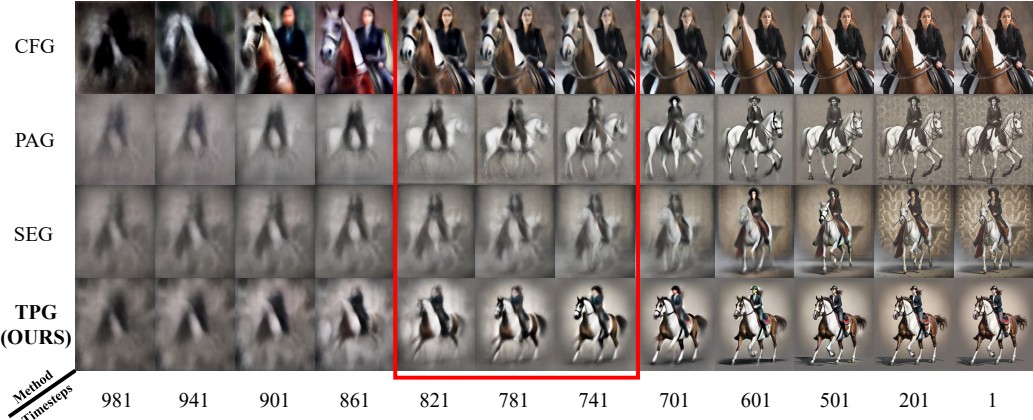

Figure 1: Visualization of the denoising process over time for different guidance strategies: CFG [17], PAG [19], SEG [18], and our proposed TPG. Each row shows generated images at various denoising time steps, from $t = 981$ (left) to $t = 1$ (right). The red box highlights the early-to-middle denoising stage ($t = 821$ to $t = 741$), where CFG and TPG demonstrate clearer structure (e.g. horse face) and consistency. The text prompt used is *"a female in a black jacket is riding a brown and white horse"*.

that CFG effectively recovers global structure and coarse details during the early denoising steps, whereas existing training-free methods tend to produce over-smoothed results at the same stage (see highlighted regions in Figure 1). This is problematic because early denoising steps are critical for both image quality and prompt alignment, as they establish global structure, major shapes, and coarse semantics before the network begins refining fine details [22, 23]. If the model fails to capture correct semantics, object placement, or overall composition at this stage, it may never fully recover from that high-level mismatch in later refinements. This lack of sufficient early-step guidance likely explains why existing methods often yield only marginal improvements in prompt alignment and generation quality compared to CFG.

Motivated by these insights, we introduce Token Perturbation Guidance (TPG), a novel method that directly perturbs intermediate token representations within the diffusion network, without requiring additional training or architectural changes. TPG employs token shuffling as the core operation to provide effective guidance signals. Specifically, token shuffling is (i) linear, (ii) preserves token norms, and (iii) disrupts local structure while maintaining global statistics. As shown in Figure 1, TPG exhibits behavior similar to CFG and, compared to other training-free methods, more faithfully recovers both global structure and fine details at early denoising stages.

We evaluate the effectiveness of TPG on both conditional and unconditional image generation using SDXL [5] and Stable Diffusion 2.1 [24]. Our results show that TPG achieves nearly a $2\times$ improvement in FID for unconditional generation compared to the SDXL baseline and also significantly outperforms existing perturbation-based guidance techniques. Furthermore, we observe that TPG closely mirrors CFG in terms of alignment and its effect on the sampling process, i.e., positively aligning with ground-truth noise in low-frequency bands, remaining largely orthogonal at other frequencies, and following a similar norm profile throughout denoising. These findings establish TPG as a general, condition-agnostic guidance method that extends CFG-like benefits to a broader class of diffusion models, including those for unconditional generation.

## 2 Related work

Score-based diffusion models [25, 1, 26, 3] reverse a forward noising process by learning the score function, the gradient of the log data density, at multiple noise levels to progressively transform pure Gaussian noise into realistic samples [26]. This principled estimator of the data distribution has outperformed previous generative modeling methods in both fidelity and mode coverage [23]. Since the score represents an explicit gradient field, the sampling trajectory can be guided by incorporating auxiliary gradients, leading to powerful guidance techniques such as classifier guidance (CG) [23] and classifier-free guidance (CFG) [17]. These guidance methods significantly enhance image quality and prompt alignment, albeit at the cost of oversaturation [27] and reduced diversity [17, 28].

Although CFG improves image fidelity and alignment with the input condition, it is inherently restricted to conditional generation. Moreover, since its guidance signal is defined as the difference between conditional and unconditional denoising outputs, CFG requires specific training procedures and its sampling trajectory can overshoot the desired conditional distribution, leading to skewed or oversimplified images [29]. Autoguidance [16] builds on CFG by introducing a deliberately weaker, also known as the "bad version" of the noise predictor, i.e., a less-trained denoiser network, to produce the guidance signal. While this avoids CFG's reliance on the unconditional score, identifying an effective "bad version" is non-trivial and still requires training and careful tuning of the model. By contrast, our method requires no additional training or architectural changes.

Recently, attention-based perturbation methods have shown promising results in improving the quality of generated images by leveraging or modifying the attention maps within the diffusion model's attention blocks. Self-Attention Guidance (SAG)[20] uses the model's own attention map to blur the denoiser input. Perturbed Attention Guidance (PAG)[19] replaces the attention map with an identity matrix to form the guidance signal, while Smoothed Energy Guidance (SEG) [18] applies Gaussian blurring to the attention maps. Although these techniques enhance image quality without additional training or auxiliary models, their impact on image quality and prompt alignment remains limited compared to CFG. In contrast, we show that TPG provides a stronger guidance signal, bridging the gap between the effectiveness of CFG and that of training-free guidance methods—both in terms of image quality and alignment with the input condition.

## 3 Background

**Diffusion Models** Denoising diffusion models generate samples from a data distribution $p_{\text{data}}(\mathbf{x})$ by reversing a gradual noising process [1, 25]. In the forward process, a clean sample $\mathbf{x}_0 \sim p_{\text{data}}$ is progressively corrupted into noise $\mathbf{x}_T$ through a stochastic differential equation (SDE):

$$d\mathbf{x} = \mathbf{f}(\mathbf{x}, t)\mathrm{d}t + g(t)\mathrm{d}\mathbf{w}, \tag{1}$$

where $\mathbf{f}(\mathbf{x}, t)$ and $g(t)$ are predefined functions representing the drift and diffusion coefficients, and $\mathrm{d}\mathbf{w}$ is an increment of the standard Wiener process. A widely adopted formulation is the variance-preserving (VP) diffusion process [3], where $\mathbf{x}_t$ gradually approaches an isotropic Gaussian as $t \to T$. A typical parameterization is $\mathbf{f}(\mathbf{x}, t) = -\frac{1}{2}\beta(t)\mathbf{x}$ and $g(t) = \sqrt{\beta(t)}$ where $\beta(t)$ defines the noise schedule. With this formulation, the denoising process is given by the following reverse-time SDE:

$$d\mathbf{x} = \left[-\frac{1}{2}\beta(t)\mathbf{x} - \beta(t)\nabla_{\mathbf{x}}\log p_t(\mathbf{x})\right]dt + \sqrt{\beta(t)}d\bar{\mathbf{w}} \tag{2}$$

where $\nabla_{\mathbf{x}}\log p_t(\mathbf{x})$ is the score function, representing the gradient of the log-density of the noisy data at time $t$. This gradient indicates the direction in which a sample should be updated to increase its likelihood w.r.t. the noisy data distribution. The unknown score function is typically approximated by a neural network $\mathbf{s}_\theta(\mathbf{x}, t)$, trained using denoising score matching [30]. Given a noisy sample $\mathbf{x}_t$ at time $t$, the network predicts the score of the marginal distribution $p_t$. Training is performed by minimizing a weighted denoising score matching loss:

$$\mathcal{L}(\theta) = \frac{1}{2}\int_0^T \beta(t)\mathbb{E}_{\mathbf{x}_0 \sim p_0}\mathbb{E}_{\mathbf{x}_t \sim p_{t|0}(\mathbf{x}_t|\mathbf{x}_0)}\big\|s_\theta(\mathbf{x}_t, t) - \nabla_{\mathbf{x}_t}\log p_{t|0}(\mathbf{x}_t \mid \mathbf{x}_0)\big\|_2^2\mathrm{d}t, \tag{3}$$

where $p_{t|0}(\mathbf{x}_t \mid \mathbf{x}_0)$ is the known Gaussian transition kernel of the forward SDE. Once trained, samples are generated by integrating the reverse SDE using $\mathbf{s}_\theta(\mathbf{x}, t) \approx \nabla_{\mathbf{x}}\log p_t(\mathbf{x})$ as an approximation to the score function. Conditional generation is enabled by training a score network that receives additional conditioning signals, e.g., class labels or text prompts, as input. In this case, the score network $\mathbf{s}_\theta(\mathbf{x}, t, c)$ approximates the conditional score $\nabla_{\mathbf{x}}\log p_t(\mathbf{x} \mid c)$.

**Guidance** Although diffusion models trained via denoising score matching have strong theoretical foundations, the score approximations are often inaccurate due to limited model capacity. As a result, unguided sampling using the learned score function tends to produce blurry and low-fidelity images, especially in complex tasks such as text-conditional generation. To address these limitations and improve generation quality, classifier-free guidance (CFG) [17] was introduced to steer the reverse diffusion trajectory toward higher quality outputs by linearly interpolating between conditional and unconditional score estimates. In general, guidance in a diffusion model can be defined as:

$$\tilde{s}_\theta(\mathbf{x}_t, c, t) = s_\theta^+(\mathbf{x}_t, c, t) + \gamma[s_\theta^+(\mathbf{x}_t, c, t) - s_\theta^-(\mathbf{x}_t, c, t)], \tag{4}$$

**Algorithm 1** Token Perturbation Guidance (TPG) for Diffusion Models

---

**Require:** Noisy input $\mathbf{x}_T \sim \mathcal{N}(0, \boldsymbol{I})$, shuffling matrices $\boldsymbol{S}_{k,t}$, set of perturbed layers $\mathcal{L}$, score function (denoiser) $s_\theta$, guidance scale $\gamma$, total time steps $T$

1: **for** $t = T, \ldots, 1$ **do**
2:      // *Forward pass without perturbation*
3:      $s_\theta^+(\mathbf{x}_t) \leftarrow s_\theta(\mathbf{x}_t, t)$
4:      // *Forward pass with perturbation*
5:      Run the network $s_\theta(\mathbf{x}_t)$ a second time with the following modification to have $s_\theta^-(\mathbf{x}_t)$:
6:          **for** each layer $k$ **do**
7:              **if** $k \in \mathcal{L}$ **then**
8:                  Apply token perturbation: $\boldsymbol{H}_k \leftarrow \boldsymbol{S}_{k,t}\boldsymbol{H}_k$
9:      // *Apply token perturbation guidance*
10:      $\tilde{s}_\theta(\mathbf{x}_t, t) \leftarrow s_\theta^+(\mathbf{x}_t) + \gamma(s_\theta^+(\mathbf{x}_t) - s_\theta^-(\mathbf{x}_t))$
11:      // *Update sample*
12:      $\mathbf{x}_{t-1} \leftarrow \text{SolverStep}(\mathbf{x}_t, \tilde{s}_\theta(\mathbf{x}_t, t))$
13: **return** $x_0$

---

where $s_\theta^+(\mathbf{x}_t, c, t)$ estimates the desired direction (typically the conditional score $s_\theta(\mathbf{x}_t, c, t)$), and $s_\theta^-(\mathbf{x}_t, c, t)$ acts as a negative score. In such guidance methods, samples are effectively pushed more toward a *positive* signal and away from a *negative* signal. In CFG, by assigning $s_\theta^-(\mathbf{x}_t, c, t) = s_\theta(\mathbf{x}_t, \varnothing, t)$, the samples are pushed away from the score of the unconditional data distribution and more strongly toward the given condition. Other guidance techniques implement $s_\theta^-(\mathbf{x}_t, c, t)$ with a less-trained diffusion model [16], or by perturbing attention maps or input pixels [19, 18, 20].

## 4 Token perturbation guidance

We next introduce Token Perturbation Guidance (TPG), a novel guidance method that directly perturbs token representations within the diffusion model. Unlike previous approaches that modify model weights or attention mechanisms, TPG operates on the intermediate token representations in the denoiser during inference. To improve sample quality, we use token shuffling as a simple choice of the perturbation, which preserves global structure while disrupting local patterns. TPG is training-free and does not require any changes to the model architecture, effectively extending the benefits of CFG to a broader class of diffusion models.

Let $\boldsymbol{H} \in \mathbb{R}^{B \times N \times C}$ denote the intermediate hidden representations, where $B$ is the batch size, $N$ is the number of tokens, and $C$ is the feature dimension per token. We apply the shuffling operator $\boldsymbol{S} \in \mathbb{R}^{N \times N}$ along the token dimension to get $\boldsymbol{H}' = \boldsymbol{S}\boldsymbol{H}$. These shuffled tokens are used to define the negative score $s_\theta^-(\mathbf{x}_t, c, t)$ in TPG. The shuffling operation satisfies the following properties:

- **Linearity**: This ensures that the perturbations can be expressed as a matrix multiplication, enabling efficient implementation within the denoising process. Thus, TPG does not add any noticeable overhead to the sampling process, and it has practically the same sampling cost as CFG.

- **Norm preservation:** The shuffling matrix is orthonormal, meaning it satisfies $\boldsymbol{S}^\top \boldsymbol{S} = \boldsymbol{I}$ and acts as a rigid rotation or permutation in the embedding space. This property guarantees that the preturbation preserves token norms, i.e., we have $||\boldsymbol{S}\boldsymbol{H}||_2 = ||\boldsymbol{H}||_2$. As a result, the magnitude of the feature representations remains unchanged, which helps maintain their statistical properties and prevents internal covariate shift [31].

Algorithm 1 summarizes the inference procedure of TPG. At each timestep, two forward passes are performed: one standard, and one with token perturbations applied at selected layers via shuffling matrices $\boldsymbol{S}_{k,t}$, unique at each time step $t$ and each layer $k$. The outputs are combined to guide the denoising trajectory toward higher-quality samples. Accordingly, TPG can be applied to both conditional and unconditional models and does not require additional training of the base model.

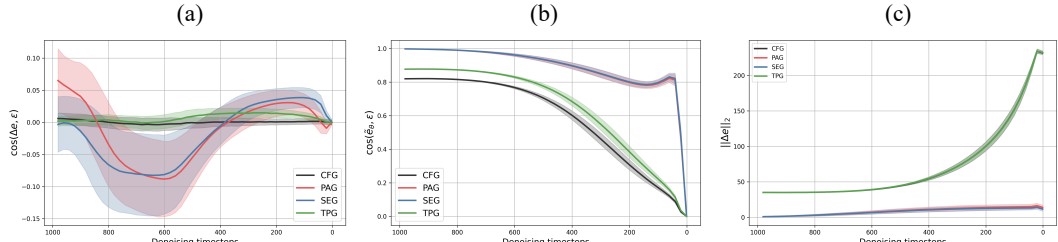

Figure 2: Analyzing the behavior of different guidance methods across denoising steps. (a) Cosine similarity between the added guidance term $\Delta e$ in $\tilde{e}_\theta = e_\theta + \gamma \Delta e$ and the true noise $\epsilon$. SEG and PAG exhibit negative alignment at intermediate steps, while TPG and CFG maintain near-zero cosine values, indicating orthogonality to the noise. (b) Cosine similarity between the full guided score $\tilde{e}_\theta$ and $\epsilon$. Compared to SEG and PAG, TPG behaves more similarly to CFG across sampling. (c) $\ell_2$ norm of the guidance term $\Delta e$. TPG and CFG follow nearly identical trends, both starting around 40 and increasing steeply in the later denoising steps. In contrast, SEG and PAG maintain consistently low norms throughout.

## 5 Comparing the behavior of TPG with other guidance methods

To better understand how different guidance strategies influence denoising, we analyzed their interaction with the true noise signal across various steps of the denoising process. In this experiment, we selected 1,000 images from the MS-COCO 2014 validation set [32]. Instead of starting from pure random noise, we corrupted each image with noise corresponding to a specific time step $t$ to generate the noisy input. This noisy image was then passed through the denoiser to produce the guided output for various methods (e.g., SEG, CFG, and TPG). We analyzed the angle between the predicted and ground-truth noise and examined the frequency components by partitioning the spectrum (up to radius 0.7) into 29 bins. Importantly, we did not use the denoiser output to progress to the next step; rather, for each time step, we reapplied noise directly to the clean image.

As shown in Figure 2, TPG and CFG produce guidance vectors that are nearly orthogonal to the ground-truth noise throughout the trajectory, as indicated by cosine values close to zero. This aligns with observation shown in [33], which demonstrates that the CFG update term can be decomposed into parallel and orthogonal components. Importantly, the parallel component primarily contributes to oversaturation. It is believed that orthogonality enables more effective steering of the predicted direction, and generally, the orthogonal component contributes to improved image quality. In contrast, PAG and SEG exhibit strong negative alignment during the middle steps, suggesting that they temporarily oppose the denoising direction. Figure 2 (b) further shows that, compared to SEG and PAG, TPG more closely mirrors the behavior of CFG in terms of alignment between the predicted and ground-truth noise. Additionally, Figure 2 (c) shows that TPG and CFG exhibit similar guidance magnitudes across all denoising steps, while SEG and PAG show substantially lower norms. This indicates that the update terms in CFG and TPG are more influential throughout the sampling process compared to that in PAG and SEG.

Figure 3 complements our step-wise analysis by illustrating how each guidance method behaves in the frequency domain. TPG and CFG remain almost perfectly orthogonal to the ground-truth noise across all frequencies and time steps, except for a slight positive tilt in the lowest frequency bands. SEG, however, exhibits a clear negative stripe at medium frequencies during intermediate steps, confirming that its correction momentarily opposes the desired direction. The norm heatmaps further reveal that CFG and TPG inject a strong low-frequency signal in the early steps, while high-frequency modifications primarily occur in the later denoising steps. In contrast, SEG operates with less energy and displays a markedly different norm pattern when modifying high-frequency content, indicating a relatively weaker approach to detail refinement. This behavior aligns with Figure 1, where the reduced energy across frequency bands leads to overly smooth generations in the initial steps.

In summary, these results show that the TPG update closely mirrors the behavior of CFG both in direction and frequency content across different denoising steps, suggesting more effective performance compared to previous training-free guidance methods like PAG and SEG.

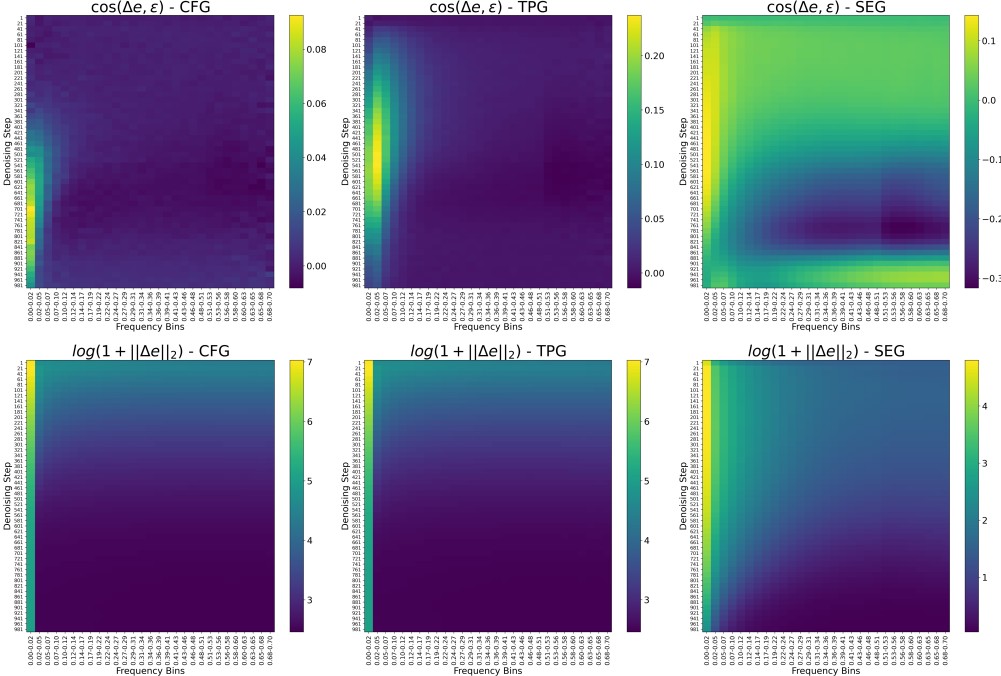

Figure 3: Frequency analysis of guidance residuals throughout sampling. Each heatmap shows either the cosine similarity between the guidance term $\Delta e$ and the ground-truth noise $\epsilon$ (top row), or the $\ell_2$ norm of the guidance term (bottom row), as a function of frequency bin (horizontal axis) and denoising step (vertical axis; $1000 \rightarrow 1$). Top: For both CFG and TPG, the guidance term remains almost orthogonal to the noise across all frequencies, with a mild positive bump in the lowest bands. In contrast, SEG transitions from weak positive alignment in the early steps to a pronounced negative stripe centered at medium frequencies. Bottom: CFG and TPG concentrate most of their energy in the lowest frequency bin and inject significantly larger magnitudes than SEG, whose energy remains up to two orders of magnitude smaller throughout the denoising process.

## 6 Experiments

**Setup**    All experiments are conducted via official implementations and pre-trained checkpoints provided by publicly available repositories. We build upon the current open-source state-of-the-art Stable Diffusion XL (SDXL) [5] as our primary baseline, and include the Stable Diffusion 2.1 (SD 2.1) [34] to show the generality of our method and analysis. Moreover, TPG is easy to implement, and it can be added into existing diffusion models as a plug-and-play module with just a few lines of additional code. In all experiments, the compared methods are evaluated using their original configurations and default guidance scales. The guidance scale for TPG is fixed at 3.0.

**Metrics**    We adopt Fréchet Inception Distance (FID) [35] as our principal metric because it correlates well with human preferences and jointly captures image quality and diversity. To quantify prompt alignment, we report the average CLIP Score [36] between sampled images and their corresponding prompts. Overall perceptual quality is further evaluated with the Inception Score and an Aesthetic Score [37]. Moreover, all experiments are conducted by generating 30k samples for each method (unless stated otherwise) and evaluated on the MS-COCO 2014 validation set [32].

### 6.1 Quantitative results

Table 1 presents a quantitative comparison of TPG against vanilla SDXL, PAG, SEG, and CFG for both unconditional and conditional image generation tasks. In unconditional generation, TPG outperforms all compared methods by achieving notably lower FID and sFID scores (69.31 and 44.18, respectively), indicating better overall image quality and diversity. Additionally, TPG achieves the highest Inception Score (17.99) while maintaining a competitive Aesthetic Score, further reflecting the perceptual quality of TPG outputs. In conditional generation, CFG achieves the best results overall, with TPG closely following and consistently outperforming vanilla SDXL, PAG, and SEG in FID, sFID, and CLIP Score metrics.

Table 1: Quantitative comparison of TPG against vanilla SDXL [5], PAG [19], SEG [18], and CFG [17] for unconditional and conditional image generation. Lower FID and sFID scores indicate superior image quality and diversity, while higher Inception, Aesthetic, and CLIP scores indicate enhanced perceptual quality and semantic alignment. TPG achieves the best metrics in unconditional generation and closely matches the performance of CFG in conditional generation.

| Setting | Metric | Vanilla SDXL [5] | PAG [19] | SEG [18] | CFG [17] | TPG (Ours) |
|---|---|---|---|---|---|---|
| Unconditional | FID↓ | 124.04 | 98.83 | 82.64 | - | **69.31** |
| | sFID↓ | 78.91 | 94.71 | 74.98 | - | **44.18** |
| | Inception Score↑ | 9.19 | 13.74 | 13.22 | - | **17.99** |
| | Aesthetic Score↑ | 5.02 | 5.94 | **6.15** | - | 6.14 |
| Conditional | FID↓ | 48.97 | 20.49 | 23.94 | **12.79** | 17.77 |
| | sFID↓ | 43.71 | 28.78 | 31.50 | **23.31** | 24.32 |
| | Inception Score↑ | 22.10 | 34.66 | 30.29 | **42.75** | 34.89 |
| | Aesthetic Score↑ | 5.37 | 6.11 | 6.18 | **6.20** | 6.12 |
| | CLIP Score↑ | 27.47 | 29.67 | 29.49 | **32.03** | 30.15 |

Table 2: Quantitative comparison of TPG with vanilla Stable Diffusion 2.1 [24], PAG [19], and SEG [18] for unconditional generation. TPG outperforms other baselines in all evaluated metrics.

| Metric | Vanilla SD [24] | PAG [19] | SEG [18] | TPG (Ours) |
|---|---|---|---|---|
| FID↓ | 25.24 | 21.30 | 20.98 | **16.69** |
| Inception Score ↑ | 24.59 | 28.80 | 25.15 | **36.28** |
| Aesthetic Score ↑ | 5.07 | 5.93 | 5.83 | **5.97** |
| Clip Score ↑ | 27.74 | 29.03 | 28.53 | **29.30** |

Table 2 further compares TPG against vanilla Stable Diffusion 2.1, PAG, and SEG in unconditional image generation. TPG consistently achieves the best performance, significantly outperforming the other methods with the lowest FID score (16.69) and highest Inception Score (36.28). It also maintains competitive results in terms of Aesthetic Score (5.97) and CLIP Score (29.30), highlighting TPG's strength in producing high-quality and semantically aligned images.

## 6.2 Qualitative results

Figure 4 highlights the differences in unconditional generations produced by various guidance methods. Vanilla SDXL, PAG, and SEG often generate abstract or repetitive textures lacking clear semantic structure, while TPG consistently produces well-structured and realistic scenes. Across various initial seeds, TPG is less prone to generating abstract patterns and more likely to form coherent spatial layouts with identifiable objects. Moreover, visual artifacts such as unnatural textures or distortions are more noticeable in outputs from baseline methods, whereas TPG reduces such artifacts, resulting in cleaner and more realistic generations.

We also show a qualitative comparison of conditional image generation across different guidance methods in Figure 5. As can be seen, CFG and TPG consistently produce the highest-quality images, with sharp details and strong alignment to the text prompts. In contrast, other baselines such as SEG and PAG often generate outputs that deviate from the given condition, leading to less faithful semantic content. Furthermore, visual artifacts, such as distorted shapes or inconsistent textures, are more frequently observed in SEG and PAG outputs, while TPG exhibits improved robustness, generating cleaner and more semantically accurate images with fewer artifacts.

## 6.3 Ablating other norm-preserving perturbation methods

To assess the importance of shuffling during guidance, Table 3 reports the results of various norm-preserving perturbation methods compared to the vanilla baseline. In addition to the shuffling strategy used in TPG, we evaluated Sign Flip, Hadamard, and Haar transforms, all described in Appendix A. While these three alternatives provide slight improvements in generation quality over the baseline, their gains are modest. In contrast, shuffling yields a substantially larger improvement in both FID and Inception Score, indicating significantly better image quality and diversity. Although all perturbations are orthogonal and preserve token norms, they influence the features differently. Shuffling randomly reorders tokens, disrupting local patterns while preserving recoverable global

| Vanilla SDXL | PAG | SEG | **TPG (Ours)** |

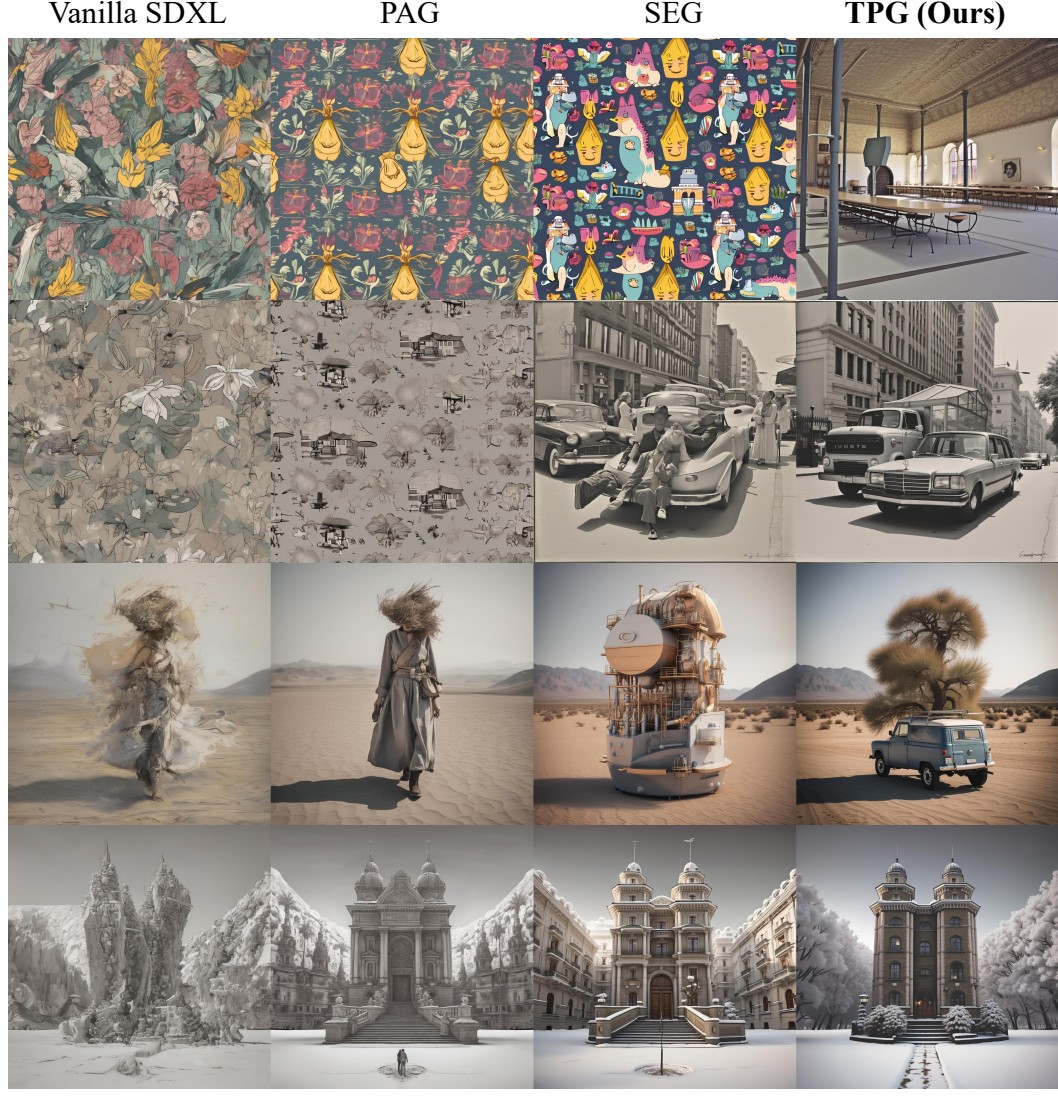

Figure 4: Qualitative comparison of unconditional generations produced by Vanilla SDXL [5], PAG [19], SEG [18], and our method (TPG). TPG achieves more realistic generations compared to other training-free guidance methods.

Table 3: Comparison of different token perturbation methods evaluated using 5K samples.

| Metric | Vanilla | Sign Flip | Hadamard | Haar | Shuffling |
|---|---|---|---|---|---|
| FID↓ | 131.57 | 119.23 | 120.54 | 118.47 | **78.43** |
| Inception Score↑ | 9.21 | 10.98 | 10.34 | 10.75 | **18.26** |

structure, which facilitates stronger guidance during inference. By comparison, Hadamard and Haar transformations mix all tokens together, potentially distorting useful information and weakening the guidance signal. Sign Flip merely alters the sign of each token, which may not offer a strong enough signal to steer samples effectively toward desirable regions of the data distribution.

## 7   Conclusion and discussion

In this paper, we introduced Token Perturbation Guidance (TPG), a novel, training-free method for enhancing the quality of diffusion models by directly perturbing intermediate token representations. TPG employs token shuffling to define an effective guidance signal, extending the benefits of classifier-

| Vanilla SDXL | CFG | PAG | SEG | **TPG (Ours)** |
|---|---|---|---|---|

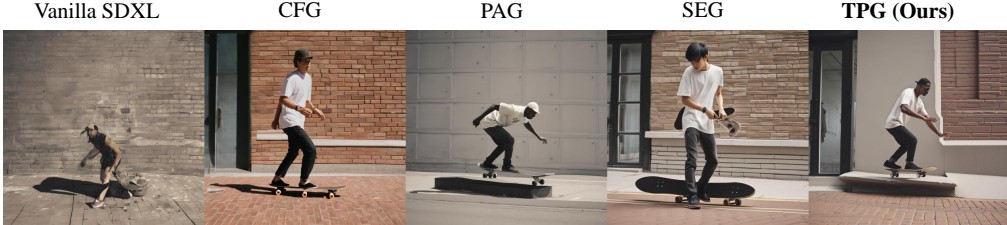

**Prompt:** "A man riding a skateboard down a brick sidewalk."

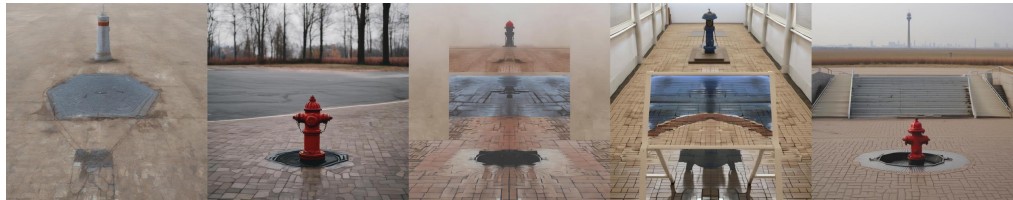

**Prompt:** "a fire hydrant in the middle of a large paved area"

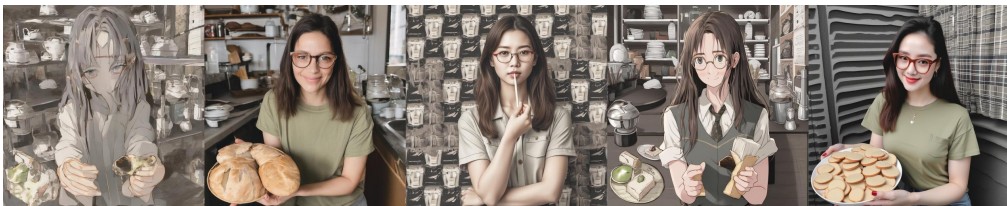

**Prompt:** "A woman with glasses that is holding a piece of bread."

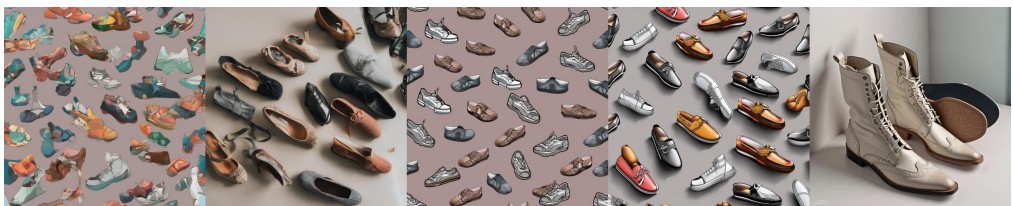

**Prompt:** "A picture of some shoes on a table."

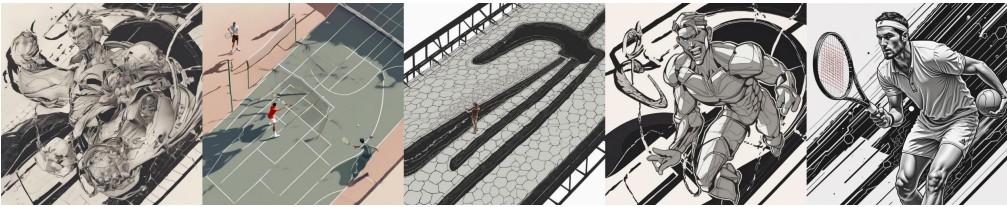

**Prompt:** "There is a man playing tennis on the court."

Figure 5: Qualitative comparison of conditional generations produced by Vanilla SDXL [5], CFG [17], PAG [19], SEG [18], and our method (TPG). TPG is able to achieve good quality and prompt alignment compared to other baselines such as PAG and SEG.

free guidance to a broader range of models. Through extensive experiments, we demonstrated that TPG improves the quality of both conditional and unconditional generation, while also enhancing prompt alignment in conditional setups. Our analysis further showed that, unlike existing attention-based perturbation methods such as SEG and PAG, the behavior of TPG closely resembles that of CFG in terms of the direction and frequency content of the guidance term. We thus consider TPG a simple, plug-and-play method that effectively bridges the gap between existing training-free guidance methods and CFG both in quality and prompt alignment.

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

# A   Orthogonal token perturbation matrix designs

TPG directly applies structured perturbations to the intermediate token embeddings within the denoiser during inference. Specifically, consider the intermediate hidden-state activations of the denoiser at a given layer, represented as a tensor $\boldsymbol{H} \in \mathbb{R}^{B \times N \times C}$, where $B$ denotes the batch size, $N$ the number of tokens, and $C$ the dimension of each token's feature vector. At each denoiser layer $k$ and diffusion time step $i$, we apply the perturbation by multiplying the token-embedding with an orthogonal (or approximately orthogonal) matrix $\boldsymbol{P}_{k,i} \in \mathbb{R}^{N \times N}$ along the token dimension:

$$\boldsymbol{H}' = \boldsymbol{P}\boldsymbol{H}.$$

The primary goal of these perturbations is to preserve global information flow while disrupting local correlations that may lead to overfitting or artifacts. To achieve this, we investigated four distinct perturbation methods for $\boldsymbol{P}_{k,i}$:

- **Token Shuffling.** Represented by a permutation matrix $\boldsymbol{S}_{k,i} \in \mathbb{R}^{N \times N}$, where $k$ denotes the denoiser's block index and $i$ the time step. The permutation matrix rearranges the tokens by selecting exactly one token from each position and assigning it to a new position; mathematically, this means that each row and column contains exactly one entry of "1", while all other entries are zeros. It simply changes the order of tokens without altering their magnitude or norm, satisfying:

$$\boldsymbol{S}_{k,i}^{\top}\boldsymbol{S}_{k,i} = I.$$

- **Random Sign Flipping.** This perturbation method is defined using a diagonal matrix $\boldsymbol{D}_{k,i} \in \mathbb{R}^{N \times N}$, whose diagonal entries $d_j$ are drawn independently and identically from $\{+1, -1\}$. Each token's embedding is thus flipped in sign independently. By construction, the matrix $\boldsymbol{D}_{k,i}$ ensures that the $\ell_2$-norm of every token embedding is preserved, by satisfying the orthogonality condition:

$$\boldsymbol{D}_{k,i}^{\top}\boldsymbol{D}_{k,i} = I.$$

- **Walsh–Hadamard Transform (WHT).** The WHT uses a normalized Hadamard matrix $\boldsymbol{W} \in \mathbb{R}^{N \times N}$, whose entries are $\pm 1/\sqrt{N}$ and which satisfies $\boldsymbol{W}^{\top}\boldsymbol{W} = I_N$. When $N$ is a power of two (i.e. $N = 2^m$), we compute the transform of an $N \times C$ token matrix $X$ in $m = \log_2(N)$ iterative stages. We begin with $\boldsymbol{W}^{(0)} = X$, and for each stage $s = 1, \ldots, m$, update

$$\begin{aligned} \boldsymbol{W}_{j,:}^{(s)} &= \boldsymbol{W}_{j,:}^{(s-1)} + \boldsymbol{W}_{j+2^{s-1},:}^{(s-1)}, \\ \boldsymbol{W}_{j+2^{s-1},:}^{(s)} &= \boldsymbol{W}_{j,:}^{(s-1)} - \boldsymbol{W}_{j+2^{s-1},:}^{(s-1)}, \end{aligned} \quad j = 1, 3, \ldots, N - 2^{s-1} + 1.$$

After $m$ stages, the result $\boldsymbol{W}^{(m)}$ equals $W X$. This structured, deterministic mixing redistributes each token's information uniformly across all others while preserving every token's $\ell_2$-norm.

- **Haar-Random Orthogonal Perturbation.** At each denoiser block $k$ and diffusion time step $i$, we generate a dense orthogonal matrix $\boldsymbol{Q}_{k,i} \in \mathbb{R}^{N \times N}$ by first sampling $\boldsymbol{A} \sim \mathcal{N}(0,1)^{N \times N}$ and then computing its QR decomposition $\boldsymbol{A} = \boldsymbol{Q}\boldsymbol{R}$. We set $\boldsymbol{Q}_{k,i} = \boldsymbol{Q}$. Since the entries of $\boldsymbol{A}$ are i.i.d. Gaussian, the orthogonal factor $\boldsymbol{Q}$ is distributed uniformly (with respect to the Haar measure) over the orthogonal group $\mathcal{O}(N)$, and by construction

$$\boldsymbol{Q}_{k,i}^{\top}\boldsymbol{Q}_{k,i} = I.$$

This yields an isotropic rotation in the $N$-dimensional token-index space, mixing all token positions globally without changing their $\ell_2$-norm.

Each method preserves the overall norm and energy of the embeddings but changes their local structure in uniquely effective ways. Empirically, these operations break up small-scale noise patterns that the denoiser might overfit, while still carrying global structure for high-quality sample generation. Among these methods, token shuffling typically provides the best overall performance: it is straightforward to implement, has minimal computational overhead, and consistently achieves significant improvements in both diversity and fidelity of generated samples.

Table 4: Effect of guidance scale $\sigma$ on generative performance. Increasing $\sigma$ improves both FID and Inception Score up to $\sigma=3$, after which performance degrades, suggesting an optimal balance between diversity and fidelity at moderate noise levels.

| $\sigma$ | FID↓ | sFID↓ | Inception Score↑ | Precision↑ | Recall↑ |
|---|---|---|---|---|---|
| 0 | 136.01 | 86.42 | 7.48 | 0.21 | 0.31 |
| 1 | 84.05 | 68.35 | 16.05 | 0.38 | 0.37 |
| 2 | 77.25 | 70.81 | 17.39 | 0.45 | 0.35 |
| 3 | **76.00** | **75.92** | **18.47** | **0.45** | **0.37** |
| 4 | 77.62 | 80.29 | 17.87 | 0.44 | 0.30 |
| 5 | 79.44 | 83.23 | 17.11 | 0.43 | 0.32 |
| 6 | 81.52 | 86.63 | 16.39 | 0.41 | 0.30 |
| 7 | 84.17 | 90.15 | 15.39 | 0.40 | 0.28 |

Table 5: Quantitative comparison of Vanilla SD3, TPG, and PAG for **unconditional generation**, evaluated over 5k generated samples. TPG achieves the lowest FID and highest Inception Score, showing improved fidelity and diversity over other baselines.

| Method | FID↓ | sFID↓ | Inception Score↑ | Precision↑ | Recall↑ |
|---|---|---|---|---|---|
| Vanilla SD3 | 113.86 | 91.09 | 11.06 | 0.26 | 0.28 |
| PAG | 138.08 | 216.65 | 9.13 | 0.25 | 0.15 |
| **TPG (Ours)** | **83.01** | **71.59** | **13.34** | **0.46** | **0.42** |

Table 6: Quantitative comparison of Vanilla SD3, CFG, TPG, and PAG for **conditional generation**, evaluated over 5k generated samples. CFG achieves the highest precision, while TPG attains a balanced trade-off between quality and diversity.

| Method | FID↓ | sFID↓ | Inception Score↑ | Precision↑ | Recall↑ |
|---|---|---|---|---|---|
| Vanilla SD3 | 33.81 | 63.17 | 29.62 | 0.45 | 0.45 |
| CFG | **21.22** | 58.14 | **42.10** | **0.70** | 0.42 |
| PAG | 41.32 | 117.03 | 25.65 | 0.42 | 0.30 |
| **TPG (Ours)** | 21.37 | **57.01** | 34.49 | 0.56 | **0.49** |

# B More ablation

**Effect of guidance scale on TPG**   We investigate the impact of the guidance scale $\sigma$ on the quality and diversity of generated samples. As shown in Table 4, increasing $\sigma$ from 0 to 3 consistently improves both FID and Inception Score, indicating better fidelity and diversity. The performance peaks at $\sigma=3$ with the lowest FID (76.00) and the highest Inception Score (18.47), suggesting an optimal trade-off between structure preservation and generative diversity. Beyond this point, larger guidance scales ($\sigma > 3$) lead to gradual degradation across all metrics, implying that excessive stochasticity introduces artifacts and reduces consistency in generation.

**Compatibility with ViT-based models**   To evaluate the generalization ability of TPG beyond the U-Net architecture, we compare it with existing perturbation-based guidance methods, including PAG, using the Stable Diffusion 3 model. The results for both unconditional and conditional generation are shown in Tables 5 and 6. Unlike prior perturbation-based approaches that were specifically designed for U-Net and fail to transfer effectively, TPG demonstrates strong cross-architecture compatibility. In the unconditional setting, TPG achieves the best overall performance, significantly reducing FID and improving Inception Score compared to both Vanilla SD3 and PAG. In the conditional setting, TPG performs competitively with CFG, achieving comparable FID and recall while maintaining high visual fidelity. These results confirm that TPG generalizes effectively across different architectures and retains its effectiveness without modification.

Table 7: Quantitative comparison of shuffling applied to down, mid, and up layers for unconditional generation using SDXL, evaluated over 5k generated samples.

| Layers | FID↓ | sFID↓ | Inception Score↑ | Precision↑ | Recall↑ |
|---|---|---|---|---|---|
| Down layers | **76.00** | **75.92** | **18.47** | **0.45** | **0.37** |
| Mid layers | 100.80 | 142.40 | 12.27 | 0.29 | 0.32 |
| Up layers | 102.28 | 154.87 | 12.81 | 0.30 | 0.23 |

Table 8: Quantitative comparison of Token Shuffling (ours), Token Blurring, and Vanilla for **unconditional generation** using SDXL, evaluated over 5k generated samples. Token Shuffling substantially improves all quality metrics, demonstrating its effectiveness in enhancing generative fidelity and diversity.

| Method | FID↓ | sFID↓ | Inception Score↑ | Precision↑ | Recall↑ |
|---|---|---|---|---|---|
| Vanilla | 136.01 | 86.42 | 7.48 | 0.21 | 0.31 |
| Token Blurring | 157.67 | 184.40 | 6.70 | 0.18 | 0.23 |
| **Token Shuffling (ours)** | **76.00** | **75.92** | **18.47** | **0.45** | **0.37** |

**Ablation on layer selection**  Following prior works such as SEG and PAG for U-Net-based models, we examined perturbing the downsampling, mid, and upsampling layers separately. Table 7 shows that TPG is most effective when the perturbation is applied only to the downsampling layers (i.e., the encoder part of the U-Net). We also experimented with combinations of down, mid, and up layers, but these did not lead to improvements and, in some cases, resulted in degraded performance.

**Exploring non-norm preserving perturbations**  We further investigate the effect of non–norm-preserving perturbations by introducing Token Blurring, which applies a Gaussian blur kernel to the input tokens. As shown in Table 8, Token Blurring leads to a clear degradation across all evaluation metrics, with higher FID and lower Inception Score compared to our Token Shuffling strategy. In contrast, Token Shuffling maintains feature magnitude consistency while introducing structured diversity, resulting in improved fidelity and perceptual quality. These results indicate that preserving the token norm during perturbation is crucial for stable and semantically coherent generation.

# C   Limitations and future work

Despite these strengths, as with CFG itself, TPG still requires two forward passes through the diffusion network, leading to increased sampling time compared to the unguided case. Additionally, although TPG significantly improves quality in most situations, the guidance term may remain limited in extreme out-of-distribution scenarios that are not captured by the learned distribution of the base model. We consider addressing these limitations an interesting direction for future research.

# D   Societal impact

Generative modeling, particularly in the domains of images and videos, holds immense potential for misuse, raising important ethical concerns. While advancements in sample quality, such as those achieved through our method, can make generated content more realistic and convincing, this heightened believability can unfortunately facilitate the spread of disinformation. Such misuse may have far-reaching negative effects on society, including the amplification of existing stereotypes and the inadvertent reinforcement of harmful biases. Although our improvements do not introduce entirely new uses for the technology, they may nonetheless increase the risk of these unintended consequences. It is therefore crucial to remain vigilant and consider the broader societal impacts that enhancements in generative modeling capabilities might entail.

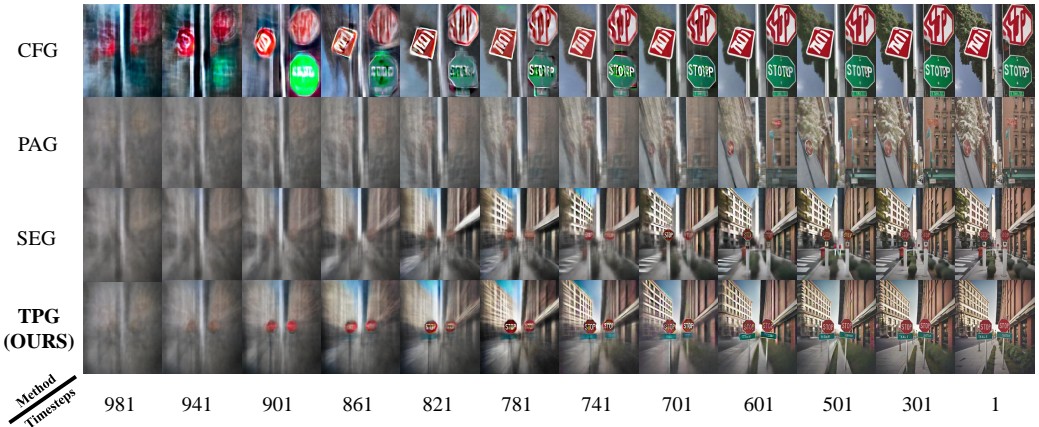

Figure 6: Visualization of the denoising process over time for different guidance strategies: CFG [17], PAG [19], SEG [18], and our proposed TPG. Each row shows generated images at various denoising time steps, from $t = 981$ (left) to $t = 1$ (right). The text prompt used is *"A red stop sign underneath green street signs"*.

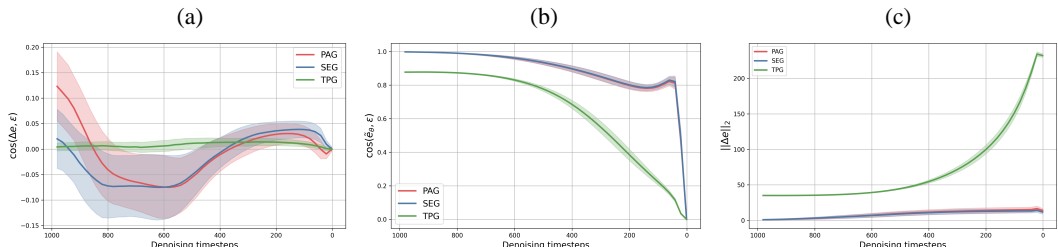

Figure 7: **Analysis of guidance behavior across denoising steps in unconditional setting. (a)** Cosine similarity between the added guidance term $\Delta e$ and the true noise $\epsilon$. **(b)** Cosine similarity between the full guided score $\tilde{e}_\theta$ and $\epsilon$. **(c)** $\ell_2$ norm of the guidance term $\Delta e$.

# E    Additional qualitative results

In this section, we provide additional qualitative results to showcase the effectiveness and adaptability of our Token Perturbation Guidance (TPG) method across different generation tasks and to compare its performance with other existing methods.

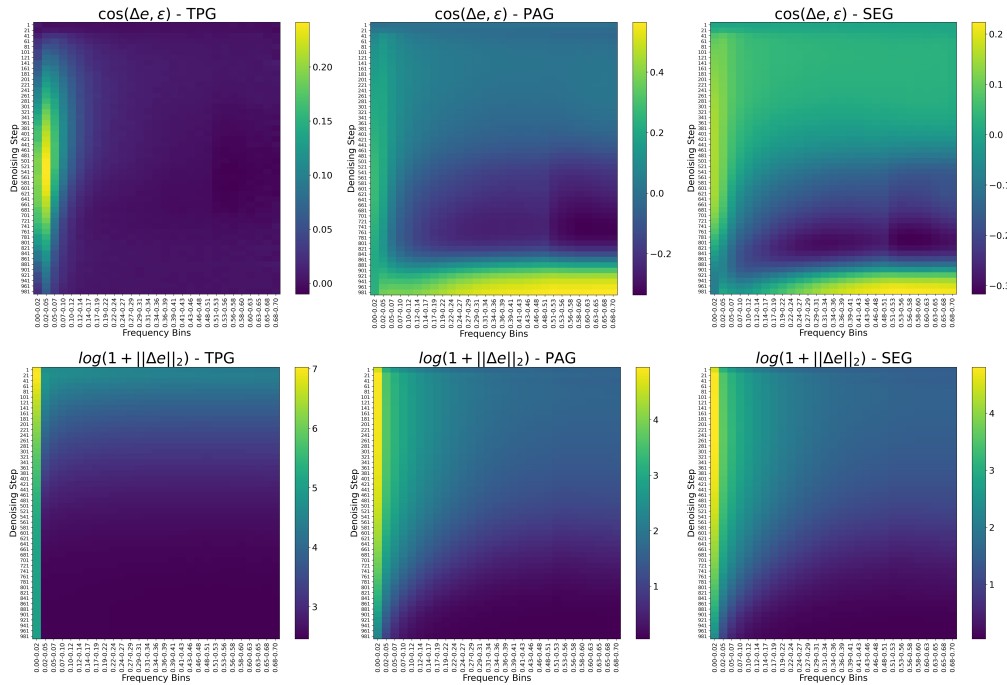

Figure 8: **Frequency–step analysis of guidance residuals in unconditional setting.** Each heat-map plots either the cosine similarity between the guidance term $\Delta e$ and the ground-truth noise $\epsilon$ (top row) or the $\ell_2$-norm of the guidance term (bottom row) as a function of frequency bin (horizontal axis) and denoising step (vertical axis; $1000 \rightarrow 1$).

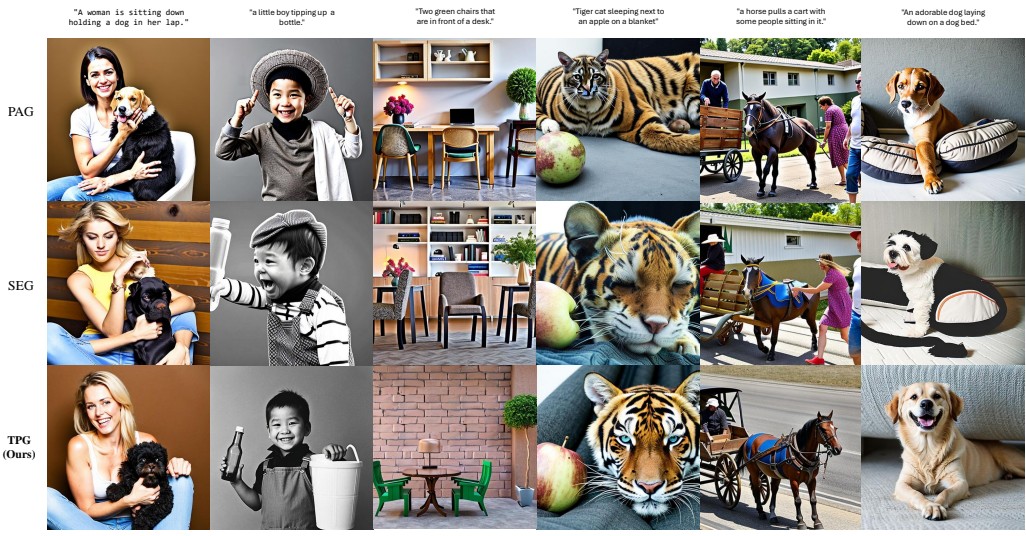

Figure 9: Qualitative comparison of conditional generation based on Stable Diffusion 2.1 [24] produced by PAG [19], SEG [18], and our TPG.

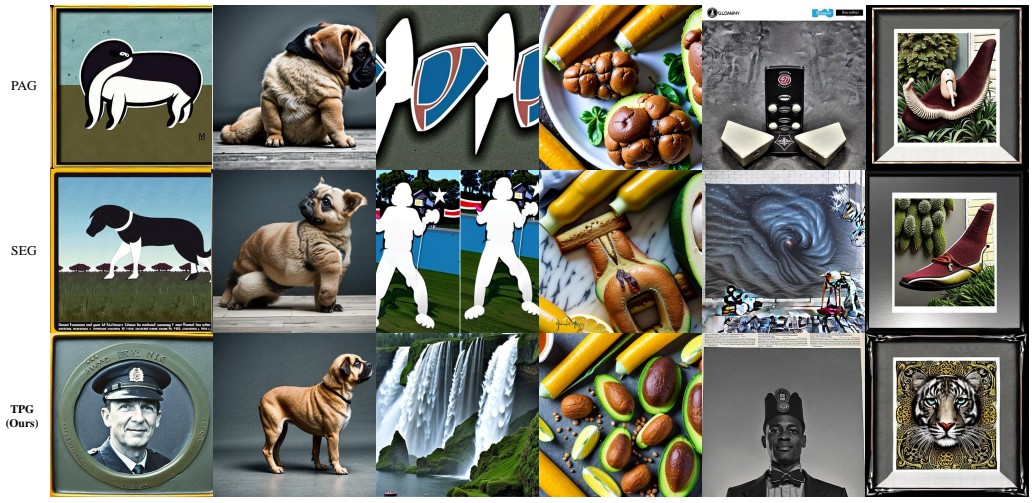

Figure 10: Qualitative comparison of unconditional generation based on Stable Diffusion 2.1 [24] produced by PAG [19], SEG [18], and our TPG.

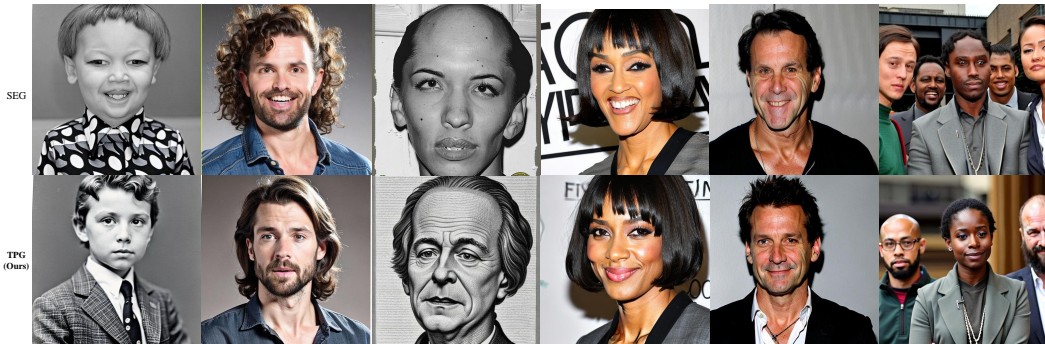

Figure 11: Qualitative comparison of face images based on Stable Diffusion 2.1 [24] generated by SEG [18] and by our TPG under both conditional and unconditional settings. SEG [18] clearly produces unrealistic patterns in the generated faces.

| Vanilla SDXL | CFG | PAG | SEG | **TPG (Ours)** |
| --- | --- | --- | --- | --- |

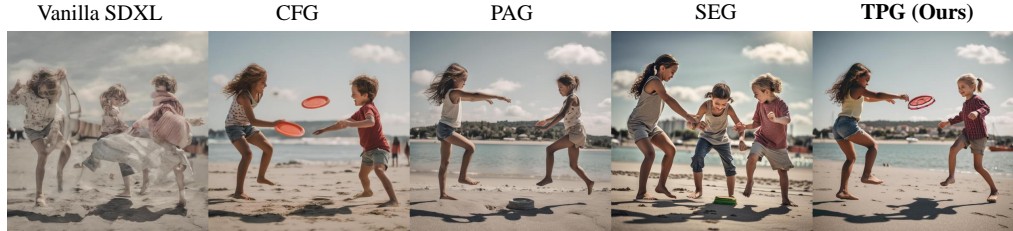

**Prompt:** "Two young children playing Frisbee on a beach."

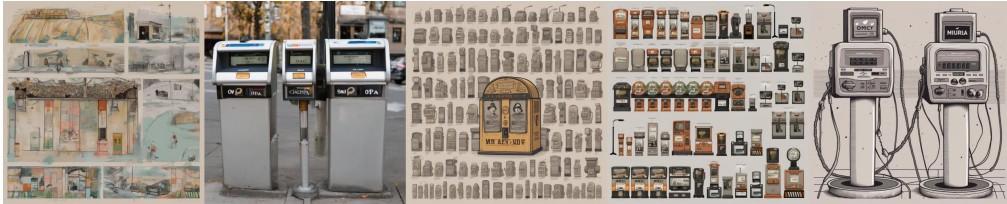

**Prompt:** "A pair of parking meters that take credit cards as well as change."

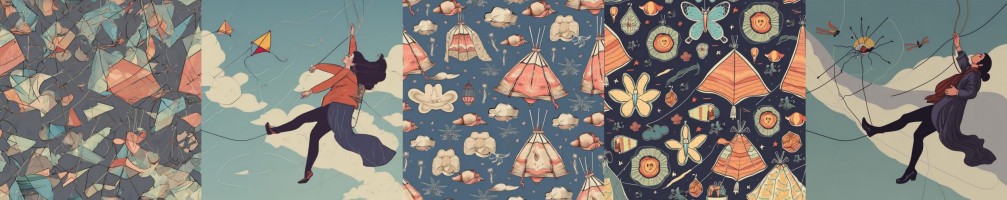

**Prompt:** "A woman flies a kite very high in the sky."

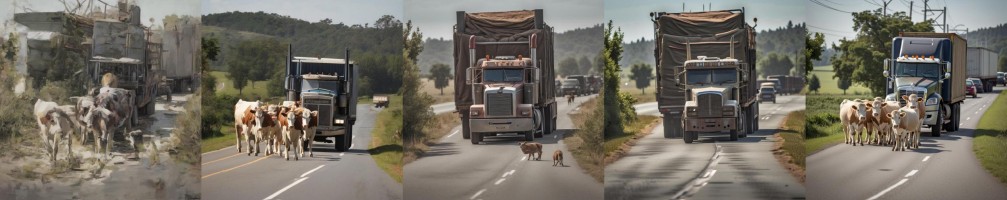

**Prompt:** "Cows crossing a country road behind a large truck."

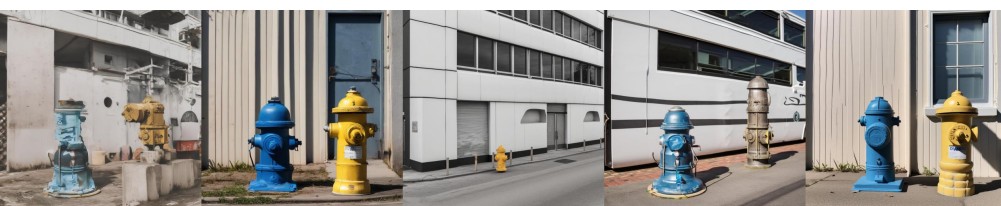

**Prompt:** "A blue and yellow fire hydrant."

Figure 12: Qualitative comparison of conditional generations produced by Vanilla SDXL [5], CFG [17], PAG [19], SEG [18], and our TPG.

Vanilla SDXL            PAG            SEG            **TPG (Ours)**

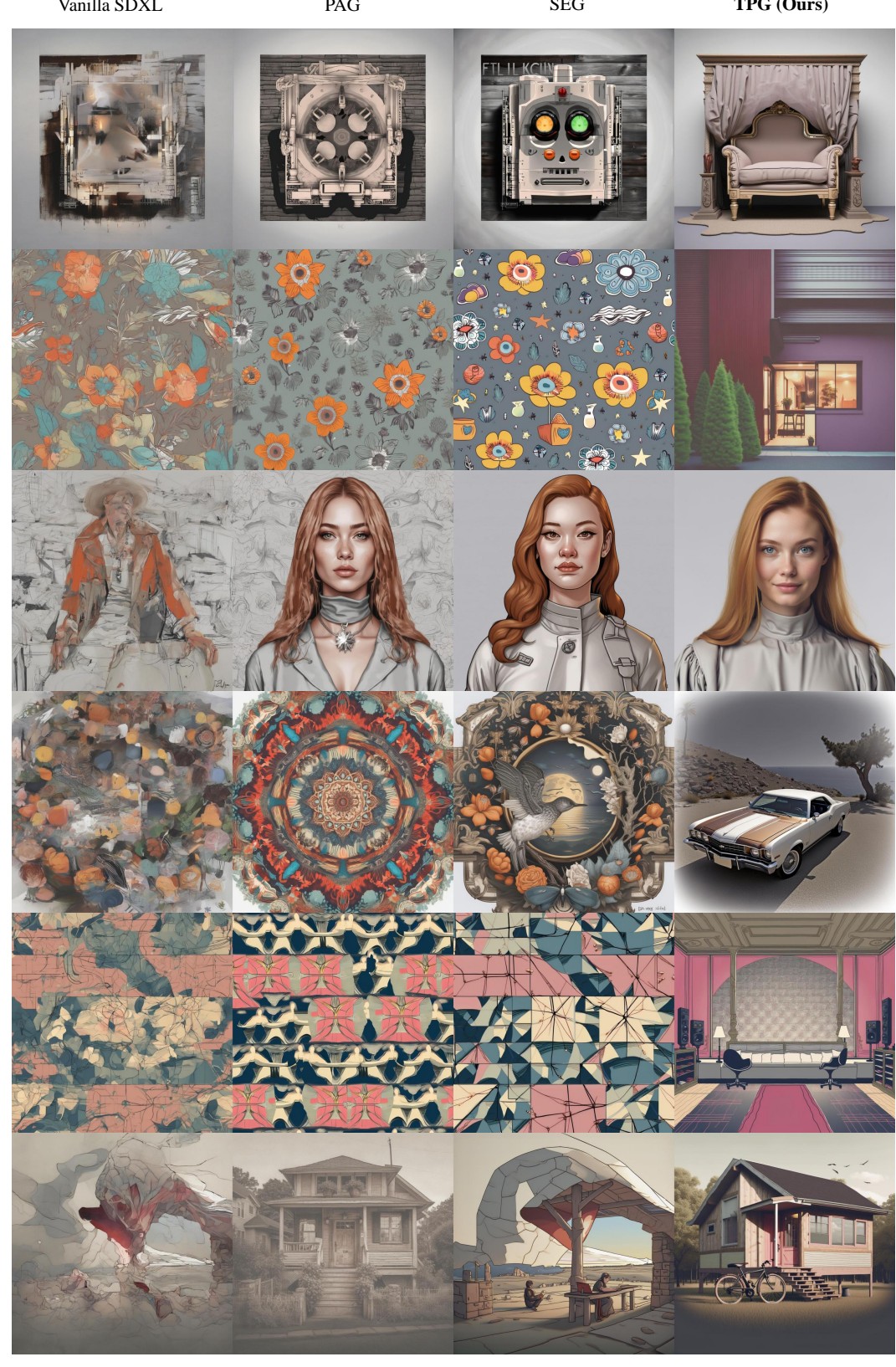

Figure 13: Qualitative comparison of unconditional generations produced by Vanilla SDXL [5], PAG [19], SEG [18], and TPG.

Vanilla SDXL          PAG          SEG          **TPG (Ours)**

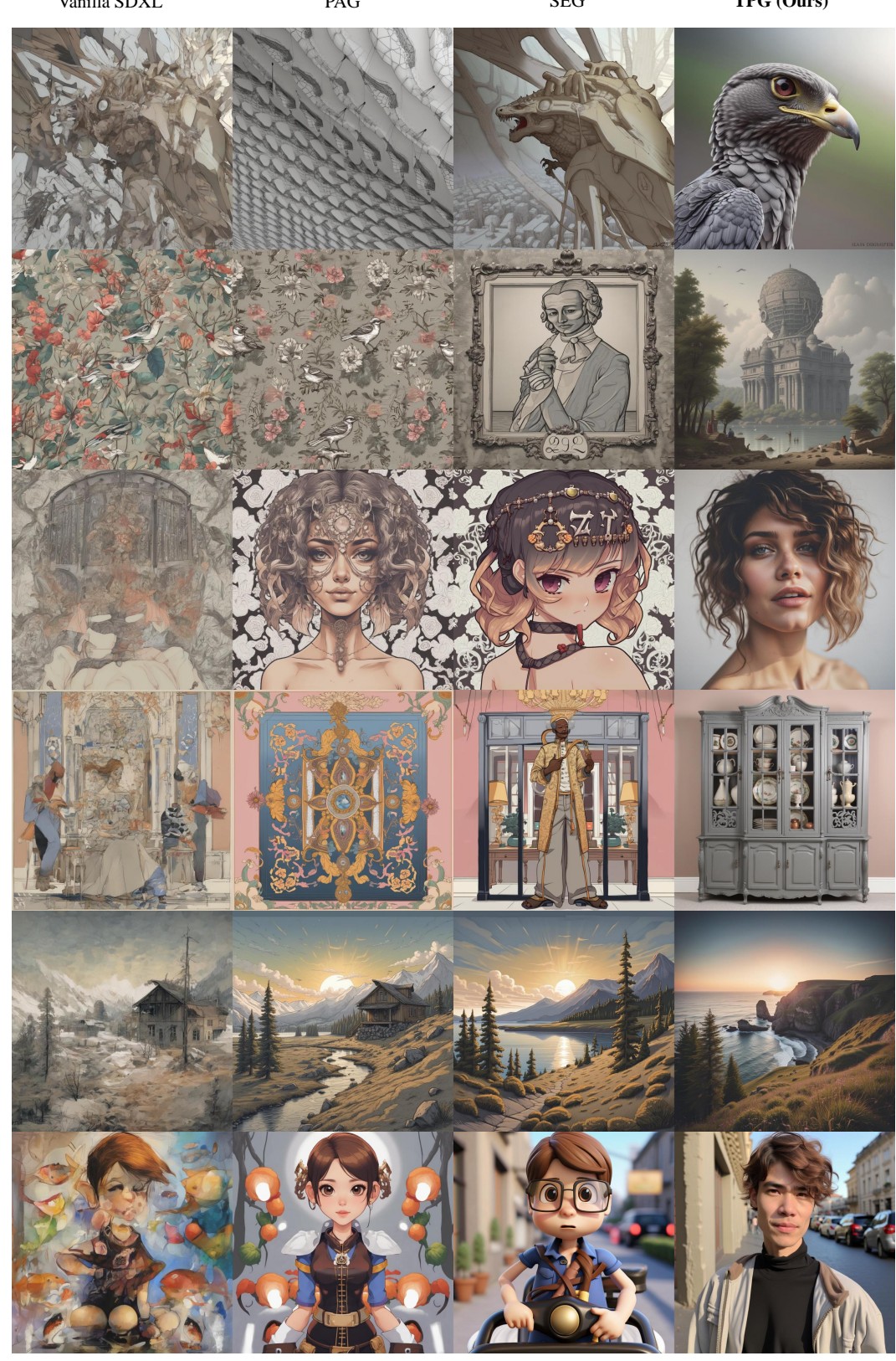

Figure 14: More qualitative comparison of unconditional generations produced by Vanilla SDXL [5], PAG [19], SEG [18], and TPG.

