# OpenReview forum: "Token Perturbation Guidance for Diffusion Models"
_NeurIPS.cc/2025/Conference — NeurIPS 2025 poster_

### Official Review · Reviewer_fVGy · 2025-06-29

**Clarity:** 3
**Significance:** 3
**Originality:** 3
**Rating:** 5
**Confidence:** 4

**Summary:**

this paper proposes a simple yet effective method to apply guidance to diffusion inference. in particular, the proposed method operates on the intermediate token representations using a shuffling operator. the experiments shows its effectiveness especially in unconditional generation scenarios.

**Questions:**

1. in the paper, S is defined as a shuffling operator, and is described as "Shuffling randomly changes the order of tokens. This breaks local patterns but still keeps the global structure recoverable" in line 218-220. but according to the statement between line 128-134, S seems not to be a transformation that is more than "changing the orders". what exactly is "permutation" and "shuffling" mean in the paper? extra analyses and visualization on the S matrix would be good.

2. why the behavior of TPG closely resembles that of CFG (as shown in Fig 2, Fig 3)?  does it imply TPG is a special case of CFG?

3. can TPG and CFG be used together? what would be the potential benefits (or problems)?

**Ethical Concerns:**

["NO or VERY MINOR ethics concerns only"]

**Final Justification:**

the rebuttal solves most of my concerns.

**Limitations:**

yes

**Quality:**

3

**Strengths And Weaknesses:**

strength:
1. new inference guidance algorithm is an important yet under-investigated topic.
2. very good writing, clear presentation which is easy to follow / to reproduce.
3. contributions of the proposed method is well justified, either in theory or in experiments.

weakness:
1. in condition generation experiments, the method does not outperform the widely-adopted cfg, in terms of generation quality or computation efficiency.

---

> ### Author Rebuttal · Authors · 2025-07-31
>
> We sincerely appreciate the reviewer’s thoughtful and valuable feedback. We have addressed all comments in detail below, and we would welcome any further discussion.
>
> ## Performance of TPG for conditional generation
> We would like to emphasize that CFG is specifically designed and trained to improve prompt alignment in conditional generation. As such, surpassing its performance in this setting is inherently challenging for general, training-free methods, such as TPG. Nevertheless, unlike CFG, which is limited to conditional generation, TPG shows superior performance over other methods in unconditional generation and significantly narrows the gap between perturbation-based guidance methods and CFG. As noted in our response to Reviewer 83Sp, this gap is nearly eliminated in the case of Stable Diffusion 3, further highlighting the strengths and generality of our pipeline. Regarding computational efficiency, as the reviewer pointed out, there is no significant difference between TPG and CFG: both require two forward passes through the diffusion model to generate guidance signals. Moreover, since the shuffling operation used in TPG is extremely lightweight, the runtime for both methods is practically identical, resulting in negligible computational overhead from TPG.
>
>
>
> ## Permutation vs. Shuffling
> We apologize for any confusion caused. In our paper, the terms “shuffling” and “permutation” refer precisely to the same operation (i.e., randomly changing the order of tokens). We will revise the paper to consistently use a single term and clarify this explicitly in the definitions. Thank you for pointing this out.
>
> ## Similarity between TPG and CFG
> While TPG and CFG exhibit some similar behaviors in the properties of their update terms (as illustrated in Fig. 2 and Fig. 3), TPG should not be viewed as a special case of CFG. The similarities primarily arise from their common ability to effectively steer predictions toward higher-quality samples. However, the underlying mechanisms of TPG and CFG differ fundamentally: CFG explicitly leverages conditional information (e.g., text prompts), whereas TPG introduces perturbations through random permutations (shuffling) in the token space. Consequently, TPG does not require any specific training and naturally extends to unconditional generation scenarios.
>
>
>
> ## Combining TPG and CFG
>
> We thank the reviewer for this insightful question. Indeed, TPG and CFG can be effectively combined, as they operate through complementary mechanisms. Following this suggestion, we conducted an additional experiment to investigate the combined effect of TPG and CFG. Our findings confirm that integrating CFG with TPG further enhances overall performance. However, it is important to note that combining these methods requires an additional forward pass through the model, resulting in improved performance at the expense of increased computational overhead. Further improvements may be possible by optimizing the weighting scheme used for combining the predictions from CFG and TPG.
>
> |    Method    | FID ↓   | sFID ↓  | Inception Score ↑ | Precision ↑ | Recall ↑ |
> |:------------:|:-------:|:-------:|:----------------:|:-----------:|:--------:|
> |    TPG       | 23.82   | 61.32   | 34.94             | 0.53        | 0.47     |
> | **TPG + CFG**| **22.93** | **62.61** | **37.48**      | **0.55**    | **0.46** |
>
> >**Table 1**: Quantitative comparison of TPG and the combined TPG + CFG approach for conditional generation using SDXL, evaluated over 5k generated samples.

---

> > ### Comment · Reviewer_fVGy · 2025-08-04
> >
> > thanks for the rebuttal, which solves most of my concerns. i will raise my score.

---

> > > ### Author Response · Authors · 2025-08-05
> > >
> > > We thank the reviewer for acknowledging our rebuttal and for their support of our work.

---

### Official Review · Reviewer_gsL2 · 2025-06-30

**Clarity:** 3
**Significance:** 3
**Originality:** 3
**Rating:** 5
**Confidence:** 3

**Summary:**

This paper introduces a training-free way to steer diffusion models by applying a norm-preserving random shuffle to token embeddings during each denoising step. Because the shuffle matrix is orthonormal, token norms stay unchanged, so the method attaches to any frozen checkpoint without retraining or architectural edits and works for both unconditional and conditional sampling. A pair of forward passes—one standard, one shuffled—produces a guidance vector whose magnitude schedule and low-frequency focus closely mirror CFG, suggesting that TPG captures a similar gradient signal while remaining condition-agnostic.

Experiments on SDXL and SD 2.1 show that TPG almost halves FID for unconditional generation and narrows the conditional gap to CFG to roughly five FID points, while clearly outperforming other training-free baselines such as PAG and SEG across FID, sFID, Inception Score, and CLIP alignment. Frequency-domain analysis confirms that the TPG guidance vector stays nearly orthogonal to ground-truth noise and tracks CFG across timesteps, and an ablation over alternative norm-preserving transforms finds random shuffling most effective. TPG still lags CFG slightly on prompt fidelity, but it extends high-quality guidance to checkpoints that lack dedicated training.

**Questions:**

N/A.

**Ethical Concerns:**

["NO or VERY MINOR ethics concerns only"]

**Final Justification:**

The authors have addressed my concerns and questions about the paper. I think that TPG proposed in this paper is an interesting and novel method. Therefore, I believe that this paper deserved to be accepted.

**Limitations:**

N/A.

**Paper Formatting Concerns:**

N/A.

**Quality:**

3

**Strengths And Weaknesses:**

Strength:
1. This paper introduces TPG, which perturbs the intermediate token embeddings by applying a norm-preserving shuffle matrix. Because the operation affects only the activations during inference, it can be inserted into any SD (or similar) checkpoint without retraining, fine-tuning, or editing the weights. The authors supply clear pseudo-code to show how the perturbation is applied.
2. On SDXL, TPG almost halves the unconditional FID and improves Inception Score from 9.2 to 18.0. In text-conditioned sampling it closes most of the gap to classifier-free guidance (CFG): FID falls from 49 to 18, while the CLIP alignment score rises from 27.5 to 30.1, matching or exceeding other training-free baselines. Similar trends hold on SD 2.1, showing the method is not tied to one backbone.
3. The paper measures cosine similarity and frequency-band alignment between the TPG guidance term and the ground-truth noise across denoising steps. The curves show that TPG stays almost orthogonal to the noise and matches the magnitude profile of CFG, whereas PAG and SEG point in the wrong direction. This supports the claim that token shuffling gives a CFG-like gradient signal.
4. Results are evaluated on 30k samples, four complementary metrics (FID, sFID, Inception Score, CLIP/Aesthetic), two model backbones, and an ablation that compares four different norm-preserving transforms.
5. Implementing TPG requires only a second forward pass and a few lines that insert the shuffle at chosen layers, so the computation and memory overhead is modest.
6. Paper is well organised and easy to follow.

Weakness:
1. Although TPG narrows the gap, it does not yet surpass CFG in conditional generation: on SDXL, conditional FID is 17.8 for TPG versus 12.8 for CFG. The paper could discuss the gap and whether larger guidance scales or mixed strategies might close it.
2. The only evidence for why a norm-preserving shuffle mimics the CFG gradient is empirical. A theoretical analysis—perhaps leveraging properties of orthonormal transforms in residual networks—would strengthen the contribution and clarify why the method work.

---

> ### Author Rebuttal · Authors · 2025-07-30
>
> We appreciate the reviewer’s recognition of the strengths of our method and are pleased that the paper was found to be well organized and easy to follow. Our detailed response is provided below, and we remain open to further discussion and questions.
>
> ## Comparison between TPG and CFG
> It is important to emphasize that CFG is specifically designed and trained to enhance alignment with prompts in conditional generation. As such, surpassing its performance in this setting is inherently challenging for general, training-free methods like TPG. Nevertheless, unlike CFG, which is restricted to conditional generation, TPG demonstrates superior performance over other methods in unconditional generation and significantly narrows the gap between perturbation-based guidance methods and CFG. Please also note that as mentioned in our response to Reviewer 83Sp, this gap is nearly eliminated in the case of Stable Diffusion 3, further highlighting the strengths of our pipeline.
>
> ## Combining TPG with CFG
> Thank you for the insightful suggestion regarding mixed strategies. Indeed, combining TPG with CFG is feasible, as they operate through complementary mechanisms and influence the model in distinct ways. Following the reviewer’s suggestion, we conducted additional experiments to evaluate the combined effect of TPG and CFG. Our results show that this combination leads to improved performance, and further gains may be achievable by optimizing the weighting scheme used for combining the predictions from CFG and TPG.
>
> |    Method    | FID ↓   | sFID ↓  | Inception Score ↑ | Precision ↑ | Recall ↑ |
> |:------------:|:-------:|:-------:|:----------------:|:-----------:|:--------:|
> |    TPG       | 23.82   | 61.32   | 34.94             | 0.53        | 0.47     |
> | **TPG + CFG**| **22.93** | **62.61** | **37.48**      | **0.55**    | **0.46** |
>
> >**Table 1**: Quantitative comparison of TPG and the combined TPG + CFG approach for conditional generation using SDXL, evaluated over 5k generated samples.
>
> ## Impact of larger guidance scale on TPG
> Empirically, we found that a guidance scale of 3.0 yields the best results and have therefore adopted it as the default value for TPG. Additionally, we conducted comprehensive evaluations across a wide range of guidance scales to investigate the effect of larger values, as suggested by the reviewer. As shown below, increasing the guidance scale beyond 3.0 did not lead to improved performance. Please also note that larger guidance scales lead to comparable performance to our optimal scale, demonstrating the robustness of TPG with respect to the guidance scale.
>
> |     σ     |  FID ↓  | sFID ↓ | Inception Score ↑ | Precision ↑ | Recall ↑ |
> |:-------:|:-------:|:------:|:----------------:|:-----------:|:--------:|
> |    3    | **76.00** | **75.92** |  **18.47**   |  **0.45**   | **0.37** |
> |    4    |  77.62  | 80.29  |      17.87       |    0.44     |   0.30   |
> |    5    |  79.44  | 83.23  |      17.11       |    0.43     |   0.32   |
> |    6    |  81.52  | 86.63  |      16.39       |    0.41     |   0.30   |
> |    7    |  84.17  | 90.15  |      15.39       |    0.40     |   0.28   |
>
> >**Table 2:** Quantitative comparison across different guidance scales (σ) for unconditional generation using SDXL, evaluated over 5k generated samples.
>
> ## Analysis of the Behavior of TPG and CFG
>
> We thank the reviewer for this question. As discussed in the paper, our primary motivation came from the observation that existing guidance perturbation methods struggle to mimic the behavior of CFG, particularly during the early denoising steps. This limits their effectiveness and results in lower image quality and weaker prompt alignment across various settings. Unlike prior approaches that perturb weights or attention modules, we chose to operate in token space by introducing guidance signals through shuffling, which disrupts local patterns without altering the global statistics of the features. That said, we acknowledge that our findings are mainly empirical, and a rigorous theoretical analysis of TPG would be a valuable direction for future work.

---

> > ### Comment · Reviewer_gsL2 · 2025-08-04
> >
> > Thanks for your response and the additional experiments. These addressed my questions and concerns. I will increase my score accordingly.

---

> > > ### Author Response · Authors · 2025-08-05
> > >
> > > We appreciate the reviewer’s positive feedback and are glad to hear that our rebuttal resolved their concerns.

---

### Official Review · Reviewer_959p · 2025-07-01

**Clarity:** 3
**Significance:** 2
**Originality:** 2
**Rating:** 5
**Confidence:** 4

**Summary:**

The authors propose a training-free, inference-time guidance procedure for improving the generation quality of diffusion models by applying unique random shuffling operations on inputs per timestep and model layer and merging the corresponidng outputs with the outputs of standard forward passes. Unlike classifier-free guidance (CFG), the proposed method does not require training with conditioning dropout and can be applied to unconditional generation. The proposed method outperforms existing unconditional inference-time guidance methods for image generation as measured by FID and Inception Score on SDXL, and in some cases is competitive with CFG for conditional generation.

**Questions:**

__Questions__

* Did the authors experiment with only performing perturbations at a subset of layers or timesteps?

* Did the authors conduct experiments with other (potentially non-norm-preserving) perturbations that destroy local information while preserving global information, e.g. repeating latents in blocks, Gaussian blur on latents, etc.?

* Can the authors comment on the potential interplay between positional encoding and the shuffle operations? In particular, can the authors clarify whether shuffling is performed before any positional encoding is applied to tokens (meaning that local/positional information is actually destroyed and not just obfuscated)?

**Ethical Concerns:**

["NO or VERY MINOR ethics concerns only"]

**Final Justification:**

The authors addressed the majority of my concerns in the rebuttal (e.g. generalization to different architectures, comparison with other perturbation methods) and committed to clarifying key aspects of the paper. As a result, I have raised my score.

**Limitations:**

Yes.

**Paper Formatting Concerns:**

None.

**Quality:**

3

**Strengths And Weaknesses:**

__Strengths__

* The proposed method appears to be very simple and can serve as a drop-in improvement for existing models

* The authors identify characteristic behaviors of CFG when compared to attention perturbation guidance methods, including stronger emergence of both high-level (structure) and low-level (detail) features early in the generation process

* The analysis of the othogonality of guidance vectors to ground truth noise is interesting. To some degree this echoes recent work (https://arxiv.org/abs/2410.02416, ICLR '25) highlighting the importance of the orthogonal component of guidance vectors for generation quality (albeit with respect to conditional prediction vectors)

* The authors demonstrate that the proposed shuffling perturbation outperforms other norm-preserving perturbations


__Weaknesses__

* The experiments presented in Figure 2 are currently explained/motivated loosely in terms of finding a guidance method that "behaves like CFG"; it would be nice to see some additional explanation of why these behaviors are desirable for the generation process. In particular, I think the authors could briefly explain why orthogonality is desirable

* The authors do not appear to mention at which blocks (k) or timesteps (i) they perform perturbations in the main body of the paper -- it seems like all layers and steps, but this could be clearer. Similarly, the description of the shuffle operation in the main body of the paper also lacks detail -- readers will likely wonder "is it a simple random permutation of all tokens, or some localized shuffling variant (as implied in line 120)?"

* Given the simplicity of the proposed approach, it is disappointing that the authors evaluate it in only two models (both Stable Diffusion variants, both trained with the same V-objective parameterization as far as I can tell, both presumably trained on similar data). For example, while it might not permit the same analysis in terms of guidance vector norms, orthogonality, etc., guidance is commonly used for masked language modeling / discrete diffusion for both images and text, and it would be interesting to see whether token perturbation guidance produces similar improvements.

* CFG is evaluated in Table 1 for conditional generation with SDXL but not in Table 2 for conditional generation with SD2.1

---

> ### Author Rebuttal · Authors · 2025-07-30
>
> We appreciate the reviewer’s thoughtful comments and their recognition of TPG’s strengths. Please find our detailed responses below, and we would be happy to engage in further discussion if needed.
>
> ## Why is orthogonality desirable?
> We thank the reviewer for raising this important point and would like to provide additional clarification, which we will also clearly highlight in the revised paper. The concept of orthogonality originates from the observations made by [1]. They demonstrated that the CFG update term can be decomposed into parallel and orthogonal components. Importantly, the parallel component primarily contributes to oversaturation, whereas the orthogonal component enhances image quality. Inspired by this insight, we analyzed the orthogonality of guidance vectors relative to the ground truth noise. Our observations indicate that both TPG and CFG maintain a near-zero cosine similarity to the ground truth noise during initial and intermediate sampling steps, demonstrating similar behavior. In contrast, SEG and PAG exhibit distinctly different alignment patterns. We discuss these results further in Section 4 of our paper, and additional analyses can be provided upon request.
>
> ## How to perform the perturbation
> We apply TPG to all time steps and down-sampling layers in the U-Net architecture, except for the initial layers, which we empirically found to degrade performance when perturbed. As such, these initial layers are excluded. To maintain consistency across architectures, we shuffle all input tokens before applying positional encoding in the transformer blocks. We will provide further details and explicitly clarify these design choices in the final version of the paper. Additionally, we have conducted an ablation study on layer selection for SDXL in the unconditional generation setting, with the results reported below.
>
>
> |                 | FID ↓   | sFID ↓  | Inception Score ↑ | Precision ↑ | Recall ↑ |
> |:---------------:|:-------:|:-------:|:-----------------:|:-----------:|:--------:|
> | **Down layers** | 76.00   | 75.92   | 18.47             | 0.45        | 0.37     |
> | **Mid layers**  | 100.80  | 142.40  | 12.27             | 0.29        | 0.32     |
> | **Up layers**   | 102.28  | 154.87  | 12.81             | 0.30        | 0.23     |
>
> >**Table 1**: Quantitative comparison of shuffling applied to down, mid, and up layers for unconditional generation using SDXL, evaluated over 5k generated samples.
>
> ## Question about evaluation and model selection
> We acknowledge the reviewer’s concern regarding our evaluation and would like to provide further clarification. Since most prior works, such as PAG and SEG, are specifically designed for U-Net–based diffusion models, we prioritized our experiments on SDXL and SD2, both well-established models, to ensure a fair experimental setup and comparability with prior work. In response to the reviewer’s suggestion, we have also evaluated the performance of TPG on Stable Diffusion 3, which employs a completely different training setup and a transformer-based architecture. The results of this comparison are reported below and will be included in the paper.  These results confirm the robustness and generality of TPG, showing that its effectiveness extends beyond any particular model or architecture.
>
> |               |    FID ↓   |   sFID ↓   | Inception Score ↑ | Precision ↑ | Recall ↑ |
> |:-------------:|:----------:|:----------:|:----------------:|:-----------:|:--------:|
> | Vanilla SD3   |  113.86    |   91.09    |      11.06       |    0.26     |   0.28   |
> | **TPG**       | **83.01**  | **71.59**  |    **13.34**     | **0.46**    | **0.42** |
> | PAG           |  138.08    |  216.65    |      9.13        |    0.25     |   0.15   |
>
> >**Table 2:** Quantitative comparison of Vanilla SD3, TPG, and PAG for unconditional generation, evaluated over 5k generated samples.
>
> |                    |    FID ↓   |   sFID ↓   | Inception Score ↑ | Precision ↑ | Recall ↑ |
> |:-------------:|:----------:|:----------:|:----------------:|:-----------:|:--------:|
> | Vanilla SD3     |  33.81  | 63.17  |      29.62       |    0.45     |   0.45   |
> | **CFG**         | **21.22** | **58.14** | **42.10**    | **0.70**    | **0.42** |
> | TPG             |  21.37  | 57.01  |      34.49       |    0.56     |   0.49   |
> | PAG             |  41.32  | 117.03 |      25.65       |    0.42     |   0.30   |
>
> >**Table 3:** Quantitative comparison of Vanilla SD3, CFG, TPG, and PAG for conditional generation, evaluated over 5k generated samples.
>
> ## Unconditional generation results in Table 2
> We thank the reviewer for raising this point. We have noticed that the mention of "conditional generation" in Table 2 is a typo; the results shown correspond to unconditional generation using SD2.1. Since unconditional generation is generally a more challenging task, our intention was to highlight the effectiveness of TPG in this setting. We will correct this in the revised version of the paper and also include results for conditional generation with SD2.1 for completeness.
>
> ## Exploring other perturbations
> Following the reviewer’s suggestion, we conducted an additional experiment to explore perturbations that are not norm-preserving. One such perturbation is applying a Gaussian blur kernel to the input tokens, which we refer to as Token Blurring. As shown in the following table, Token Blurring demonstrates suboptimal performance compared to Token Shuffling in TPG. We present the results of this perturbation below, and we will include these findings in the paper.
>
> |           | FID ↓   | sFID ↓  | Inception Score ↑ | Precision ↑ | Recall ↑ |
> |:---------:|:-------:|:-------:|:----------------:|:-----------:|:--------:|
> | Vanilla   | 136.01  | 86.42   | 7.48             | 0.21        | 0.31     |
> | Token Blurring | 157.67 | 184.40 | 6.70          | 0.18        | 0.23     |
> | **Token Shuffling (ours)** | **76.00** | **75.92** | **18.47**  | **0.45** | **0.37** |
>
> >**Table 4**: Quantitative comparison of Token Shuffling (ours), Token Blurring, and Vanilla for unconditional generation using SDXL, evaluated over 5k generated samples.
>
>
> ------
> >[1] Sadat S, Hilliges O, Weber RM. Eliminating oversaturation and artifacts of high guidance scales in diffusion models. In The Thirteenth International Conference on Learning Representations 2024 Oct 3.

---

> > ### Comment · Reviewer_959p · 2025-08-05
> > **Reply to Authors**
> >
> > I thank the authors for their detailed rebuttal.
> >
> > Re: orthogonality, I suspected that the paper from Sadat et al. was an inspiration and hence mentioned it in my review. Given that the authors confirm this to be the case in their rebuttal, I am confused as to why they do not appear to mention or cite it in their paper. The authors should probably fix this.
> >
> > The clarification and additional experiment on layer selection for TPG is good to see. The choice of layers definitely needs to be mentioned clearly in the main body of the paper, and the additional experiment can probably go in an appendix if space is short.
> >
> > The results for SD3 are encouraging and will improve the paper.
> >
> > I also appreciate the authors running the experiment with non-norm-preserving perturbations, which seems to indicate that shuffling is a good choice of perturbation.
> >
> > One thing is still not clear to me: is the shuffle operation a global random shuffle, or some local variant? Again, line 121 of the paper seems to imply the latter, but the paper and rebuttal do not seem to mention if this is the case.
> >
> > Provided the authors rectify the missing citation and clarify the shuffling mechanism, I think a majority of my concerns have been addressed and I am willing to raise my score.

---

> > > ### Author Response · Authors · 2025-08-05
> > >
> > > We appreciate the reviewer’s positive feedback and are glad that our rebuttal addressed most of their concerns.
> > >
> > > We'll make sure to add the missing citation to the paper and further clarify the importance of orthogonality.
> > >
> > > Regarding the shuffling operation, it refers to a global random shuffle (i.e., randomly changing the order of tokens). In line 121, we meant that the content of each token remains unchanged, while the spatial relationships between neighboring tokens are disrupted by the shuffling.
> > >
> > > We will also include the provided experiments and clarifications in the final version of the paper.

---

### Official Review · Reviewer_83Sp · 2025-07-06

**Clarity:** 2
**Significance:** 2
**Originality:** 2
**Rating:** 2
**Confidence:** 5

**Summary:**

This paper introduces Token Perturbation Guidance (TPG), a new method designed to improve the generation quality and alignment of diffusion models. TPG addresses limitations of Classifier-Free Guidance (CFG), which requires specific training and is restricted to conditional generation. TPG works by applying a norm-preserving shuffling operation directly to intermediate token representations within the diffusion network. Authors examine their method in various experiments to show the effectiveness of the proposed methods.

**Questions:**

1. This paper lacks sufficient novelty and can be considered a simplified version of citation [18]. Moreover, the proposed method in [18] can lead to a semantically different model, whereas the method proposed in this paper is overly simplistic and may fail to bring any fundamental improvement. The proposed method in [18] should also considered as a baseline.

2. The method does not adapt well to the ViT architecture. For instance, before and after the perturbation, the relationships between features are essentially only affected through the positional encoding, which may be functionally similar to simply shuffling the input. In fact, this should be included and discussed as a baseline.

3. The paper should include a discussion on the impact of the perturbation scale. From the current perspective, the proposed method may lead to instability in the output, depending on how the perturbation is applied.

4. The paper should further quantify the benefits of the proposed method during the "early-to-middle" training stages, rather than presenting only a single figure.

5. The method shows only marginal improvements. The performance gap between the conditional setting and CFG should be analyzed and explained in detail.

6. The paper does not provide a reasonable strategy for determining which feature layers (L) should be perturbed. This omission may lead to significant manual tuning effort.

**Ethical Concerns:**

["NO or VERY MINOR ethics concerns only"]

**Final Justification:**

Thanks for the explanation. I still believe that this paper is a simple version of existing method, therefore I keep my score unchanged.

**Limitations:**

yes

**Quality:**

2

**Strengths And Weaknesses:**

Strengths
1. Well-organized and Clear Writing.

2. Interesting Idea.

Weaknesses
1. Limited Novelty: The method appears to be a "vanilla" version of the method proposed in [18].

2. Unreasonable methodology: The proposed method is overly simplistic and straightforward, lacking a clear analysis of how it works and how it differs from other simple baselines.

3. Limited Performance Improvement: The experimental results show only modest performance gains.

---

> ### Author Rebuttal · Authors · 2025-07-30
>
> We would first like to thank the reviewer for finding our proposed method to be an interesting idea and for acknowledging the well-organized and clear writing of our paper. We would like to address the concerns and questions raised as follows:
>
> ## Comparison with AutoGuidance
> AutoGuidance [1] uses a smaller/weaker version of the same conditional model for guidance. This requires training a separate model with reduced capacity and shorter training time, as well as accessing two models during sampling. In contrast, our method involves no additional training or architectural modifications. Since a weaker version of Stable Diffusion is not publicly available, a direct comparison between our method and AutoGuidance is not feasible. Training such a model would exceed our computational resources, requiring several weeks on multiple high-end GPUs. On the other hand, our method is training-free and can be readily applied on top of any pretrained model, which we consider an important advantage of TPG over AutoGuidance.
>
> ## Simplicity of TPG
> We acknowledge the simplicity of our proposed method, as the reviewer noted. However, we view this straightforwardness as a key strength rather than a limitation. TPG offers a simple and easy-to-implement adaptation for existing diffusion models that is condition-agnostic and requires no additional training. It consistently improves generation quality across a range of setups, including both conditional and unconditional generation. Notably, we believe that one of the main reasons for the popularity and widespread adoption of classifier-free guidance is its ease of use. While CFG is limited to conditional generation and requires a specific training setup, TPG can be seen as a similarly practical approach that extends these benefits to a broader class of diffusion models.
>
> ## Analysis on why TPG works
> As discussed in the paper, our main motivation came from the observation that other guidance perturbation methods fail to mimic the behavior of CFG, especially during the early denoising steps. This limits their effectiveness and results in lower image quality and weaker prompt alignment across various setups. Unlike previous approaches that operate in weight space or attention modules, we chose to operate in token space by forming guidance signals through shuffling, which disrupts local patterns without altering the global statistics of the features. One of our most important and interesting findings is that both TPG and CFG maintain a near-zero cosine similarity with the ground truth noise during the initial and intermediate steps. This aligns with the concept of orthogonality, previously introduced in [2]. It is believed that orthogonality enables more effective steering of the predicted direction, and generally, the orthogonal component contributes to improved image quality. We will clarify this point further in the final version of the paper.
>
> ## Concerns about marginal improvements
> We have shown that TPG not only narrows the gap between CFG and other training-free methods in conditional generation but also achieves state-of-the-art performance in the unconditional setting. As illustrated in Figure 4, other methods often fail to produce semantically coherent images, whereas TPG consistently generates results with higher visual quality and semantic consistency. This is further supported by our quantitative evaluations, where TPG outperforms other training-free guidance baselines by a noticeable margin.  For example, Table 2 shows TPG achieves an FID of 69.31 compared to 124.04 for vanilla SDXL, which is nearly a 2$\times$ improvement. Taken together, we consider the improvements achieved by using TPG to be both fundamental and significant.
>
> ## Compatibility with ViT-based models
> We would like to emphasize that our method is not limited by model architecture and is *compatible* with ViT-based models (e.g., Stable Diffusion 3) as well. While previous perturbation-based guidance methods (e.g., PAG and SEG) were specifically designed for the U-Net architecture and may not generalize well to other backbones, we found that this limitation does not apply to TPG. To demonstrate this, we conducted an experiment comparing TPG with other baselines using Stable Diffusion 3, and report results for both conditional and unconditional generation below.
>
>
> |               |    FID ↓   |   sFID ↓   | Inception Score ↑ | Precision ↑ | Recall ↑ |
> |:-------------:|:----------:|:----------:|:----------------:|:-----------:|:--------:|
> | Vanilla SD3   |  113.86    |   91.09    |      11.06       |    0.26     |   0.28   |
> | **TPG**       | **83.01**  | **71.59**  |    **13.34**     | **0.46**    | **0.42** |
> | PAG           |  138.08    |  216.65    |      9.13        |    0.25     |   0.15   |
>
> >**Table 1:** Quantitative comparison of Vanilla SD3, TPG, and PAG for unconditional generation, evaluated over 5k generated samples.
>
> |                    |    FID ↓   |   sFID ↓   | Inception Score ↑ | Precision ↑ | Recall ↑ |
> |:-------------:|:----------:|:----------:|:----------------:|:-----------:|:--------:|
> | Vanilla SD3     |  33.81  | 63.17  |      29.62       |    0.45     |   0.45   |
> | **CFG**         | **21.22** | **58.14** | **42.10**    | **0.70**    | **0.42** |
> | TPG             |  21.37  | 57.01  |      34.49       |    0.56     |   0.49   |
> | PAG             |  41.32  | 117.03 |      25.65       |    0.42     |   0.30   |
>
> >**Table 2:** Quantitative comparison of Vanilla SD3, CFG, TPG, and PAG for conditional generation, evaluated over 5k generated samples.
>
> This experiment shows that TPG not only achieves the highest quality in unconditional generation, but also performs nearly on par with CFG in the conditional setting. These results highlight the robustness of our method and confirm that TPG is not restricted to a single architecture. We will include this experiment in the final version of the paper.
>
> ## Ablation on the guidance scale used in TPG
> We assume the reviewer’s question refers to the guidance scale used in TPG, as our pipeline does not include any perturbation parameter. To address this, we have conducted a quantitative comparison across different guidance scales, and the results are given below:
>
> |         |  FID ↓  | sFID ↓ | Inception Score ↑ | Precision ↑ | Recall ↑ |
> |:-------:|:-------:|:------:|:----------------:|:-----------:|:--------:|
> | σ = 0     | 136.01 |	86.42| 7.48| 0.21 | 0.31
> | σ = 1   |  84.05  | 68.35  |      16.05       |    0.38     |   0.37   |
> | σ = 2   |  77.25  | 70.81  |      17.39       |    0.45     |   0.35   |
> | σ = 3   | **76.00** | **75.92** |  **18.47**   |  **0.45**   | **0.37** |
> | σ = 4   |  77.62  | 80.29  |      17.87       |    0.44     |   0.30   |
> | σ = 5   |  79.44  | 83.23  |      17.11       |    0.43     |   0.32   |
> | σ = 6   |  81.52  | 86.63  |      16.39       |    0.41     |   0.30   |
> | σ = 7   |  84.17  | 90.15  |      15.39       |    0.40     |   0.28   |
>
> >**Table 3:** Quantitative comparison across different guidance scales (σ) for unconditional generation using SDXL, evaluated over 5k generated samples.
>
> We will include this experiment as an ablation study in the final paper. Although a scale of 3 yields the best performance, TPG is robust to this choice, with other values also achieving reasonable results relative to the baseline ($\sigma=0$).
>
> ## Quantifying the benefits of early-to-middle denoising steps
> To the best of our knowledge, there is no established metric that directly quantifies the impact of early-to-middle denoising steps in the sampling process. However, we believe these steps play a crucial role in shaping the semantics of the generated images. As such, improvements in FID and CLIP scores may be interpreted as indirect evidence of effective denoising in early sampling stages. As shown in Tables 1 and 2, our method achieves the best FID and CLIP scores in both unconditional and conditional generation settings compared with other perturbation-based guidance methods. This likely reflects the benefits of enhanced early-to-middle denoising and aligns with the behavior observed in CFG. Additional results are provided in the Appendix and more can be added upon request if needed.
>
> ## Ablation study for layer selection
> Following prior works such as SEG and PAG for U-Net-based models, we tested perturbing the downsampling, mid, and upsampling layers separately. Our initial experiments showed that TPG is most effective when the perturbation is applied only to the downsampling layers (i.e., the encoder part of the U-Net). We also experimented with combinations of down, mid, and up layers, but these did not lead to improvements and, in some cases, resulted in degraded performance. In response to the reviewer’s question, we have conducted an ablation study on layer selection for SDXL in the unconditional generation setting. The results are provided below and will be included in the final version of the paper.
>
> |              |   FID ↓   |   sFID ↓   | Inception Score ↑ | Precision ↑ | Recall ↑ |
> |:------------:|:---------:|:----------:|:----------------:|:-----------:|:--------:|
> | Down layers  | **76.00** | **75.92**  |    **18.47**     |  **0.45**   | **0.37** |
> | Mid layers   |  100.80   |  142.40    |      12.27       |    0.29     |   0.32   |
> | Up layers    |  102.28   |  154.87    |      12.81       |    0.30     |   0.23   |
>
> >**Table 4:** Quantitative comparison of shuffling applied to down, mid, and up layers for unconditional generation using SDXL, evaluated over 5k generated samples.
>
>
> -------------
> >[1] Karras T, Aittala M, Kynkäänniemi T, Lehtinen J, Aila T, Laine S. Guiding a diffusion model with a bad version of itself. Advances in Neural Information Processing Systems. 2024 Dec 16;37:52996-3021.
> [2] Sadat S, Hilliges O, Weber RM. Eliminating oversaturation and artifacts of high guidance scales in diffusion models. In The Thirteenth International Conference on Learning Representations 2024 Oct 3.

---

> > ### Author Response · Authors · 2025-08-07
> > **Gentle reminder**
> >
> > We thank the reviewer once again for their time and effort in reviewing our paper. We hope that our response has satisfactorily addressed their concerns and will assist in their final assessment of our work. We remain open to discussion should any further questions remain.

---

> > > ### Comment · Reviewer_83Sp · 2025-08-09
> > >
> > > Thanks for the explanation and the additional experiments. I have no further questions. I will update my score based on the situation.

---

### Decision · Program_Chairs · 2025-09-17

**Decision:**

Accept (poster)

**Comment:**

This paper proposes a token perturbation training-free guidance to apply a norm-preserving shuffling operation on tokens within the diffusion model to improve the generation quality. Experiments are conducted using SDXL and compare the proposed guidance with classifier-free guidance and also other related methods.

Reviewers commented positively on the writing quality, simple plug-in method, and the meaningful insight into the guidance behaviour of diffusion models. Reviewers queried about the novelty compared to similar works, the limited performance gain, the evaluation on limited baseline models, and the lower results for conditional generation. Most of the concerns have been well addressed in the rebuttal and using the supplemented results.

After the rebuttal, the remaining concern is about the distinction between the proposed method and an existing paper. Although they share some similarity in high-level ideas, the proposed method has satisfactory novelty given its different design and training-free property. This is the main reason for acceptance.